# Quantitative intra-Golgi transport and organization data suggest the stable compartment nature of the Golgi

**Hieng Chiong Tie[1,2†], Haiyun Wang[1,2†], Divyanshu Mahajan[1], Hilbert Yuen In Lam[1], Xiuping Sun[1], Bing Chen[1], Yuguang Mu[1], Lei Lu[1]***

[1]School of Biological Sciences, Nanyang Technological University, Singapore, Singapore; [2]Medisix Therapeutics, Singapore, Singapore

## eLife Assessment

This **important** study offers **convincing** evidence that intra-Golgi transport slows from cis to trans and varies between cargos even within the same cisternae, supporting a more stable compartment model. Using nocodazole-induced ministacks, the authors show cargo-specific transport kinetics with distinct velocities and residence times. These findings refine the cisternal progression model and prompt further investigation into alternative mechanisms, such as rapid partitioning or rim progression. This study will be of interest to cell biologists studying membrane trafficking, Golgi organization, and protein secretion, as well as researchers investigating the mechanisms of organelle dynamics and the molecular basis of intracellular transport.

***For correspondence:**
lulei@ntu.edu.sg

[†]These authors contributed equally to this work

**Abstract** How the intra-Golgi secretory transport works remains a mystery. The cisternal progression and the stable compartment models have been proposed and are under debate. Classic cisternal progression model posits that both the intra-Golgi transport and Golgi exit of secretory cargos should occur at a constant velocity dictated by the cisternal progression; furthermore, COPI-mediated intra-Golgi retrograde transport is essential for maintaining the Golgi organization. Leveraging our recently developed Golgi imaging tools in nocodazole-induced Golgi ministacks, we found that the intra-Golgi transport velocity of a secretory cargo decreases during their transition from the *cis* to the *trans*-side of the Golgi, and different cargos exhibit distinct velocities even within the same cisternae. We observed a vast variation in the Golgi residence times of different cargos. Remarkably, truncation of the luminal domain causes the Golgi residence time of Tac — a standard transmembrane secretory cargo without intra-Golgi recycling signals — to extend from 16 min to a notable 3.4 hr. Additionally, when COPI-mediated intra-Golgi retrograde transport was inhibited by brefeldin A, we found that nocodazole-induced Golgi can remain stacked for over 30–60 min. Therefore, our findings challenge the classical cisternal progression model and suggest the stable compartment nature of the Golgi.

## Introduction

The Golgi complex in mammalian cells plays a crucial role in membrane trafficking and post-translational modification of proteins and lipids (cargos). Centrally positioned around the microtubule organization center (*Glick and Luini, 2011*; *Klumperman, 2011*), it comprises laterally connected Golgi stacks, each containing 4–7 tightly adjacent membrane sacs known as cisternae. Conventionally, a Golgi stack is recognized to have three regions: the *cis*, medial, and *trans*-Golgi. The *trans*-Golgi network (TGN) assembles outside the *trans*-side of a Golgi stack and is mainly composed of tubular and vesicular

membranes (*De Matteis and Luini, 2008*). In the secretory pathway, newly synthesized cargos exit the ER at the ER exit site (ERES), pass the ER and Golgi intermediate compartment (ERGIC), and reach the Golgi. Cargos subsequently transit from the *cis*-side through the medial and eventually reach the *trans*-side of the Golgi, where they exit the Golgi and target the plasma membrane (*Bergmann and Singer, 1983*; *Castle et al., 1972*).

Despite decades of research, the Golgi remains one of the most enigmatic organelles, particularly regarding the mechanism of intra-Golgi transport of secretory cargos (*Emr et al., 2009*; *Glick and Luini, 2011*). Two primary intra-Golgi transport models have been proposed and are currently the subject of debate. In the classic cisternal progression or maturation model, secretory cargos passively reside within Golgi cisternae, which progress or mature from the *cis* to the *trans*-Golgi cisternae to facilitate the forward or anterograde transport of cargos. Simultaneously, post-translational modification enzymes such as glycosyltransferases move in reverse or retrograde transport, altering cisternal properties to become the next cisternae of the Golgi stack. Under this perspective, the Golgi functions at a dynamic equilibrium of anterograde and retrograde intra-Golgi transport. The model predicts that all secretory cargos should have the same and constant intra-Golgi transport and Golgi exit velocity. It can provide a plausible explanation for the intra-Golgi transition of oversized secretory cargos, such as procollagen I (*Bonfanti et al., 1998*; *Mironov et al., 2001*). Moreover, direct observations of cisternal maturation have been made in budding yeast *Saccharomyces cerevisiae* (*Kurokawa et al., 2019*; *Losev et al., 2006*; *Matsuura-Tokita et al., 2006*), although similar observations have not been reported in mammalian cells. However, the budding yeast Golgi differs significantly from the mammalian one in the cisternal organization – it scatters throughout the cytoplasm as unstacked compartments. This substantial difference casts doubt on the general applicability of the budding yeast Golgi observation to higher eukaryotes.

In contrast, the stable compartment model posits that Golgi cisternae are stable entities. During intra-Golgi transport, carriers actively move secretory cargos from one cisterna or compartment to the next, from the *cis* to the *trans*-side, while post-translational modification enzymes remain stationary. At the *trans*-side of the Golgi, cargos are sorted into carriers bound for the plasma membrane. As a result, different cargos can exhibit distinct intra-Golgi transport and Golgi exit velocities under this model. To account for the intra-Golgi transport of oversized secretory cargos, such as procollagen I, a modified version of the stable compartment model, the rim progression model, has been proposed (*Lavieu et al., 2013*; *Pfeffer, 2010*; *Volchuk et al., 2000*). In this model, each Golgi cisterna seems to have a stable interior domain and a dynamic rim domain; the rim domain of Golgi cisternae undergoes constant en bloc fission and fusion, carrying small or large cargos to the next cisternae for intra-Golgi cargo transport. The rim partitioning of large secretory cargos is supported by previous EM studies of procollagen I (*Bonfanti et al., 1998*), FM4 aggregates (*Lavieu et al., 2013*), and algal protein aggregates (*Engel et al., 2015*). Furthermore, we have recently demonstrated in fluorescence microscopy that during their intra-Golgi transition, small cargos, such as CD59, E-cadherin, and VSVG, localize to the cisternal interior, coinciding with Golgi glycosyltransferases, while large cargos, such as FM4 aggregates and collagenX, position themselves at the cisternal rim, where trafficking machinery components localize (*Tie et al., 2018*).

Once secretory cargos transit the Golgi stack, they reach the *trans*-side of the Golgi and are packed into carriers destined for the plasma membrane. The classic cisternal progression model posits that secretory cargos linearly depart the *trans*-Golgi. Contrary to this, cargo exit kinetics have been shown to adhere to a first-order exponential relationship rather than a linear one (*Hirschberg et al., 1998*; *Patterson et al., 2008*; *Sun et al., 2020*). In an attempt to reconcile these observations, *Patterson et al., 2008* introduced the rapid partitioning model, which stands distinct from the classic cisternal progression and stable compartment models. This model suggests that cargos rapidly diffuse throughout the Golgi stack, segregating into multiple post-translational processing and export domains, where cargos are packed into carriers bound for the plasma membrane. Nonetheless, synchronized traffic waves have been observed through various techniques, including electron microscopy (EM) (*Trucco et al., 2004*) and advanced light microscopy methods we developed, such as GLIM and side-averaging (*Tie et al., 2016*; *Tie et al., 2022*). These findings suggest that the rapid partitioning model might not accurately represent the true nature of the intra-Golgi transport.

The ongoing debate among these models underscores the complexity of the Golgi and reflects the insufficiency of the existing experimental data, especially the intra-Golgi transport kinetics data.

Two semi-quantitative approaches exist for studying the intra-Golgi transport kinetics of ER-synchronized secretory cargos. The first is the biochemical approach, where intra-Golgi transport kinetics can be indirectly deduced by subtracting ER-to-Golgi and Golgi-to-plasma membrane transport from the overall secretion (ER-to-plasma membrane transport) during the chase (*Ernst et al., 2018*). However, this indirect approach provides only an averaged intra-Golgi transport kinetics, making it incapable of measuring the instantaneous intra-Golgi transport velocity at a sub-Golgi region. The second approach involves EM imaging of immuno-gold-labeled secretory cargos at Golgi cisternae (*Beznoussenko et al., 2014*; *Trucco et al., 2004*). The distribution of gold particles among individual Golgi cisternae during the chase directly indicates intra-Golgi transport kinetics. Nevertheless, this approach demands specialized techniques and equipment and considerable manual work in EM imaging and subsequent image examination, limiting its widespread use in other labs.

The limitations in current approaches call for novel methods, especially ones based on fluorescence microscopy, to resolve the intra-Golgi transport spatially and kinetically. However, optically resolving the intra-Golgi secretory transport in mammalian cells is challenging due to the thinness (200–400 nm) and random orientation of Golgi stacks. To overcome this, we leverage the uniform and rotationally symmetrical arrangement of nocodazole-induced Golgi ministacks (hereafter referred to as Golgi ministacks). Extensive studies have provided strong evidence that ministacks obtained under prolonged nocodazole treatment (≥3 hr), the condition employed in our studies, largely represent the native Golgi stack (*Cole et al., 1996*; *Fourrière et al., 2016*; *Rogalski et al., 1984*; *Schueder et al., 2024*; *Tie et al., 2018*; *Trucco et al., 2004*; *Van De Moortele et al., 1993*). We developed a numerical Golgi localization tool called Golgi localization by imaging centers of mass (GLIM) that can precisely pinpoint a Golgi protein's cisternal localization with nanometer accuracy in Golgi ministacks (*Tie et al., 2017*; *Tie et al., 2016*). Alongside this, we propose using the metric of Golgi residence time to quantify a Golgi protein's retention in the Golgi (*Sun et al., 2021*; *Sun et al., 2020*).

Here, we utilized past and newly acquired GLIM and Golgi residence time data to quantitatively analyze intra-Golgi transport and Golgi exit kinetics. Our data revealed that neither intra-Golgi transport nor Golgi exit exhibits a constant velocity. We discovered that when the luminal domain of Tac — a conventional transmembrane secretory cargo lacking intra-Golgi recycling signals — is truncated, its Golgi residence time increases from 16 min to a substantial 3.4 hr. Through GLIM, we examined

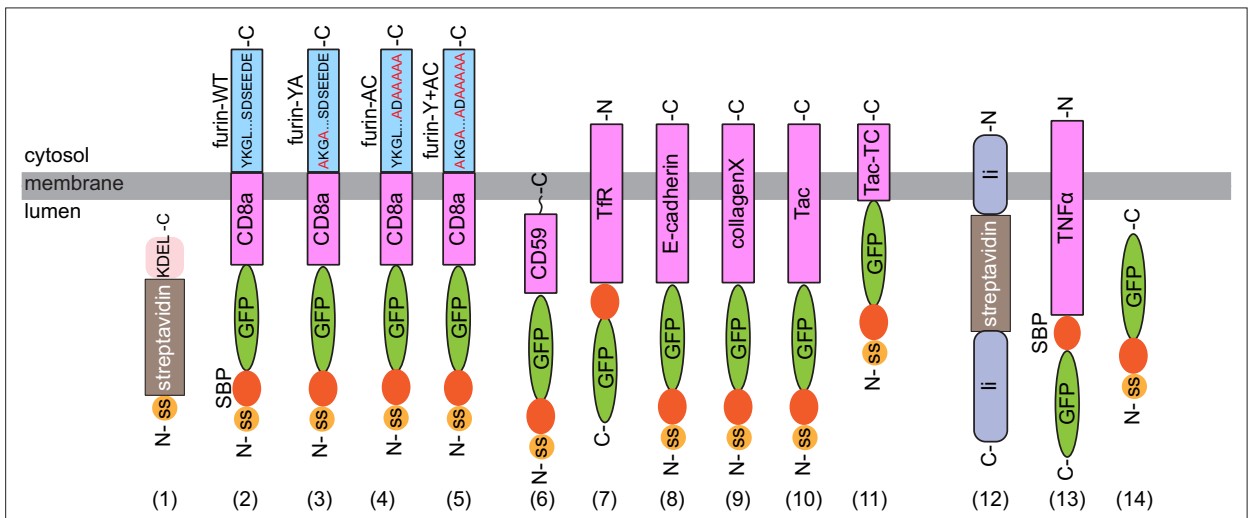

**Figure 1.** A schematic diagram showing the domain organizations of ER hooks and RUSH reporters used in this study. (**1**) Streptavidin-KDEL (ER hook), (**2**) SBP-GFP-CD8a-furin-WT, (**3**) SBP-GFP-CD8a-furin-YA, (**4**) SBP-GFP-CD8a-furin-AC, (**5**) SBP-GFP-CD8a-furin-Y+AC, (**6**) SBP-GFP-CD59, (**7**) TfR-SBP-GFP, (**8**) SBP-GFP-E-cadherin, (**9**) SBP-GFP-collagenX, (**10**) SBP-GFP-Tac, (**11**) SBP-GFP-Tac-TC, (**12**) Ii-streptavidin (ER hook), (**13**) TNFα-SBP-GFP, and (**14**) SBP-GFP. ss, signal sequence. SBP, streptavidin binding peptide. (1) is the ER hook for RUSH reporters (2-11), while (12) is the ER Hook for the RUSH reporter (13-14). Key cytosolic amino acid motifs are indicated for furin cytosolic tail RUSH reporters. YA is YKGL, a tyrosine-based motif, to AKGL mutation, while AC is SDSEEDE, an acidic cluster sequence, to ADAAAAA mutation (*Tie et al., 2016*). Y+AC indicates mutations in both sites (*Tie et al., 2016*).

The online version of this article includes the following figure supplement(s) for figure 1:

**Figure supplement 1.** Golgi localization by imaging centers of mass (GLIM) and the Auto-GLIM tool.

the cisternal organization of Golgi ministacks under brefeldin A (BFA) treatment, which halts retro-grade intra-Golgi transport. Remarkably, under this condition, we found that nocodazole-induced Golgi ministacks can remain stacked for 30–60 min. Therefore, our findings underscore the stable compartment nature of Golgi cisternae. They challenge the classical cisternal progression model and favor the stable compartment model.

## Results

### The intra-Golgi transport is not a motion with constant velocity

The classic cisternal progression model postulates that the intra-Golgi transport velocity of a secretory cargo should remain constant. However, this prediction has not been directly tested due to the inherent challenge of measuring the intra-Golgi transport velocity. The advent of GLIM has allowed us to address this issue. In GLIM, briefly, HeLa cells expressing the RUSH secretory cargo (RUSH reporter) (*Boncompain et al., 2012*) and GalT-mCherry, a *trans*-Golgi marker containing amino acids 1–81 of B4GALT1, were initially treated with nocodazole. Subsequently, cells were chased in biotin with nocodazole and cycloheximide (a protein synthesis inhibitor) for various lengths of time (*t*) before immunofluorescence labeling for endogenous GM130, a *cis*-Golgi marker. Additionally, apart from SBP-GFP, a soluble secretory protein, and SBP-GFP-CD59 (*Tie et al., 2016*), a glycosylphosphatidylinositol-anchored protein, all RUSH reporters in this study are transmembrane proteins, including TNFα-SBP-GFP, TfR-SBP-GFP (*Chen et al., 2017*), SBP-GFP-CD8a-furin-WT (*Tie et al., 2016*), SBP-GFP-CD8a-furin-YA, SBP-GFP-CD8a-furin-AC (*Tie et al., 2016*), SBP-GFP-CD8a-furin-Y+AC (*Tie et al., 2016*), SBP-GFP-collagenX (*Fourriere et al., 2019*), SBP-GFP-Tac (*Sun et al., 2020*), SBP-GFP-Tac-TC (*Sun et al., 2020*), and SBP-GFP-E-cadherin (*Boncompain et al., 2012*). Except for TNFα-SBP-GFP and SBP-GFP, which uses Ii-streptavidin as the ER hook, all RUSH reporters employ signal sequence fused streptavidin-KDEL as their ER hook (*Figure 1*).

We calculated the centers of mass as the positions of GM130, RUSH reporter, and GalT-mCherry within each analyzable Golgi ministack. The RUSH reporter's Golgi localization quotient, or *LQ*, is calculated by dividing its distance from GM130 by GalT-mCherry's distance from GM130 (*Figure 1—figure supplement 1A–G*). The *LQ* is a linear numerical metric to indicate a cargo's axial localization within the Golgi, with a nanometer range of precision (*Tie et al., 2017*; *Tie et al., 2016*). We previously linearly defined regions of Golgi: ERES/ERGIC (*LQ*<–0.25), *cis* (–0.25≤*LQ*< 0.25), medial (0.25≤*LQ*<0.75), *trans*-Golgi (0.75≤*LQ*<1.25), and TGN (*LQ*≥1.25).

To analyze the intra-Golgi transport kinetics of secretory cargos, we measured the *LQ*s of RUSH reporters after various durations of biotin administration (chase). Most kinetic data were previously reported in HeLa cells (*Tie et al., 2017*; *Tie et al., 2016*) and re-analyzed here. Additionally, we generated new data and replicated specific measurements. GLIM involves laborious manual image analysis. To increase image analysis efficiency, we developed a software tool, Auto-GLIM, to automatically analyze ministack images and calculate *LQ*s by using a deep learning algorithm (the manuscript will be published elsewhere). We found that Auto-GLIM can produce *LQ*s similar to the conventional manual analysis method (*Figure 1—figure supplement 1H–M*) but requires much less user interaction, therefore, increasing our image analysis efficiency. We employed the Auto-GLIM tool to analyze the intra-Golgi transport of RUSH reporters, SBP-GFP-CD59 and SBP-GFP-Tac-TC, in HEK293T cells.

*Figure 2*, *Figure 2—figure supplement 1* illustrate that our LQ vs. t plots are highly reproducible. As previously reported, all LQ vs. t plots fit the following first-order exponential function (*Equation 1*) well with an adjusted R$^2$ (adj. R$^2$)≥0.85 (*Figure 2*, *Figure 2—figure supplement 1A-T*, left panels).

$$LQ = y_0 - Ae^{\left(\frac{-ln2}{t_{intra}}t\right)} \tag{1}$$

In *Equation 1*, *t* represents chase time in minutes (biotin treatment starts at *t*=0); *ln2* is the natural logarithm of 2; *A* is a constant; $y_0$ represents the *LQ* of the Golgi exit site; $t_{intra}$ is the time that the cargo reaches half of the transport range and is hereafter referred to as the intra-Golgi transport time. *Table 1* lists $t_{intra}$, $y_0$, and adj. $R^2$ of each data set. We define the instantaneous intra-Golgi transport velocity as the derivative of *LQ* with respect to time, *dLQ/dt*, which measures the axial transport velocity of the center of mass of the synchronized cargo wave. It should also follow the first-order exponential function (*Equation 2*).

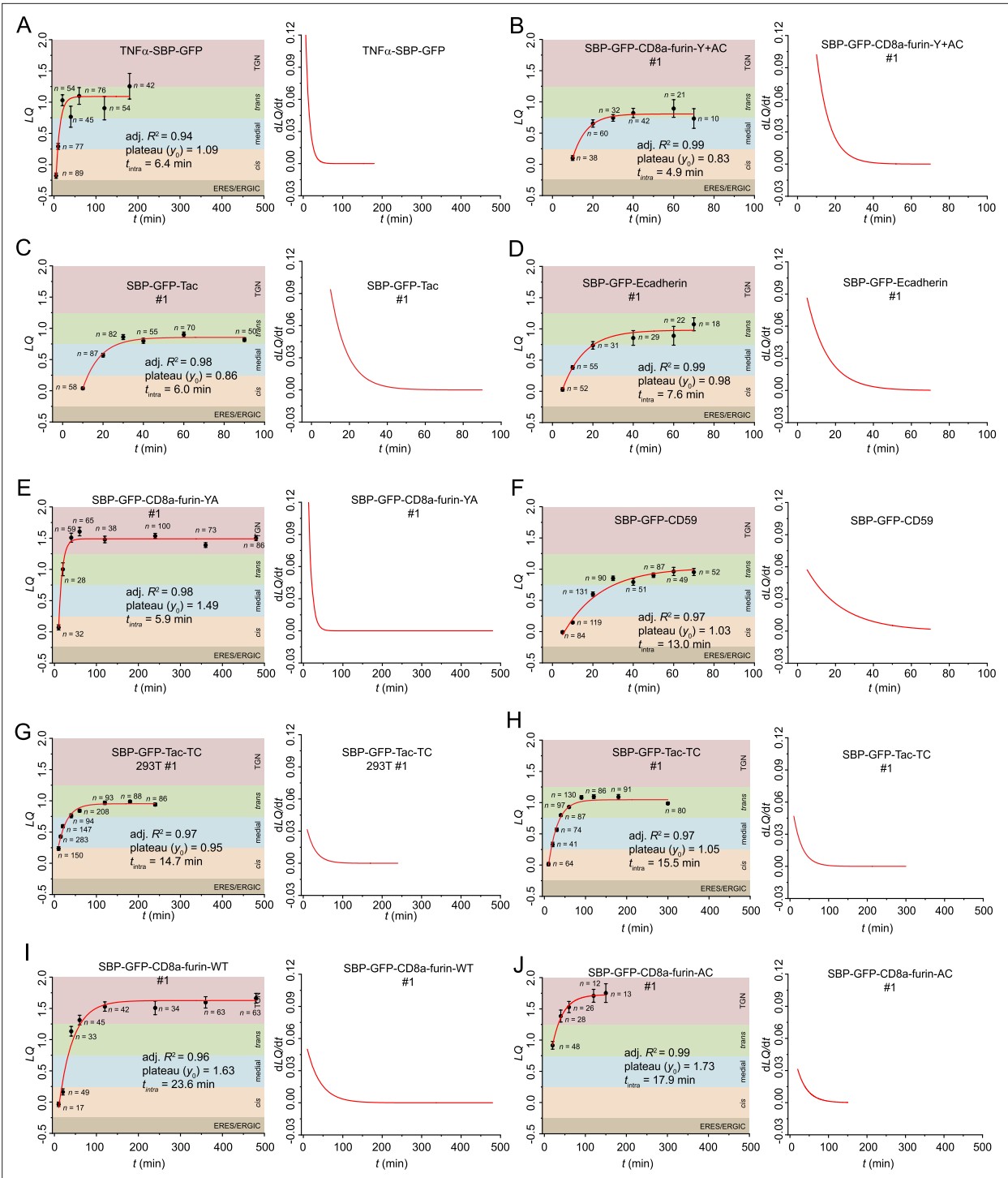

**Figure 2.** The intra-Golgi transport kinetics for RUSH Reporters in Golgi ministacks. HeLa or 293T cells transiently co-expressing individual RUSH reporter and GalT-mCherry were incubated with nocodazole for 3 hr. This was followed by a chase with nocodazole, cycloheximide, and biotin for variable time durations (**t**) before fixation and GM130 immunostaining. Except for TNFα-SBP-GFP, which uses Ii-streptavidin as the ER hook, all RUSH reporters employ signal sequence fused streptavidin-KDEL as their ER hook. Using Golgi localization by imaging centers of mass (GLIM), $LQ$s for the RUSH reporters were then calculated and plotted. Each panel has an $LQ$ vs. $t$ plot on the left, with $n$ (the number of analyzable Golgi ministacks), adj. $R^2$, $y_0$, and $t_{intra}$ indicated. Error bar, SEM. On the right side of each panel, the d$LQ$/d$t$ vs. $t$ plot represents the first-order derivative of its corresponding $LQ$ vs. $t$ plot on the left. ERES/ERGIC ($LQ<-0.25$), cis ($-0.25 \leq LQ < 0.25$), medial ($0.25 \leq LQ < 0.75$), trans-Golgi ($0.75 \leq LQ < 1.25$), and TGN ($1.25 \leq LQ$) zones were color-shaded. The $LQ$ vs. $t$ plot in panel G was acquired in 293T cells and analyzed by the Auto-GLIM tool, while the rest are from our previous reports (see **Table 1**). Panels are arranged according to their RUSH reporters' $t_{intra}$ means (see **Table 1**). See **Figure 2-source data 1** for the raw data.

*Figure 2 continued on next page*

*Figure 2 continued*

The online version of this article includes the following source data and figure supplement(s) for figure 2:

**Source data 1.** *LQ* data employed in plots presented in *Figure 2*.

**Figure supplement 1.** Additional intra-Golgi transport kinetics of RUSH reporters.

**Figure supplement 1—source data 1.** *LQ* data employed in plots presented in *Figure 2—figure supplement 1*.

$$\frac{dLQ}{dt} = \frac{Aln2}{t_{intra}}e^{\left(\frac{-ln2}{t_{intra}}t\right)} \tag{2}$$

In this context, the intra-Golgi transport velocity is the highest when the cargo enters the secretory pathway ($t=0$). However, it is crucial to approach this extrapolation cautiously due to the lack of experimental data at $t \leq 5$ min, when a RUSH reporter's high ER background and low Golgi signal make it challenging to select analyzable Golgi ministacks. It is evident that the $dLQ/dt$ of all our RUSH reporters slows to zero as they transit across the Golgi stack to reach $LQ = y_0$ at the *trans*-Golgi or TGN (*Figure 2A-J*, *Figure 2—figure supplement 1A-T*, right panels). At the *trans*-Golgi, we propose that RUSH reporters that do not target the TGN exit Golgi ministacks in carriers en route to the plasma membrane (*Tie et al., 2018*; *Tie et al., 2016*; *Tie et al., 2022*). Hence, the intra-Golgi transport velocity of a secretory cargo does not remain constant, contradicting the prediction of the classic cisternal progression model.

## The intra-Golgi transport kinetics of collagenX, a cisternal rim partitioned secretory cargo, resemble those of conventional cargos

CollagenX has been known to assemble into oligomers (*Kwan et al., 1991*). We previously reported that the collagenX RUSH reporter, SBP-GFP-collagenX, forms large aggregates containing ~190 copies (*Tie et al., 2018*). With an estimated mean size of ~40 nm, these aggregates are much smaller than FM4 aggregates and procollagen I (>300 nm) (*Bonfanti et al., 1998*; *Volchuk et al., 2000*) and, therefore, are not excluded from conventional transport vesicles, which typically have a size of 50–100 nm. However, our previous findings showed that while conventional secretory cargos partition to the cisternal interior during intra-Golgi transport, large cargos such as FM4 aggregates and collagenX preferentially localize to the cisternal rim (*Tie et al., 2018*), highlighting distinct intra-Golgi transport behavior for different cargo sizes. Hence, we asked if collagenX follows the same intra-Golgi transport kinetics as conventional secretory cargos.

Using the RUSH assay, our *LQ* vs. *t* data demonstrated that the intra-Golgi transport of SBP-GFP-collagenX followed a first-order exponential function, with a $t_{intra}$ of 8±1 (mean ± SD, n=3) (*Figure 3A–C*; *Table 1*). SBP-GFP-collagenX exited at the *trans*-Golgi, with a $y_0$ of 0.92±0.08 (mean ± SD, n=3). Additionally, the instantaneous intra-Golgi transport velocity ($dLQ/dt$) of SBP-GFP-collagenX also decreases in accordance with a first-order exponential function (*Figure 3A–C*; *Table 1*).

Using side averaging, we visualized the synchronized traffic wave of SBP-GFP-collagenX as it transitioned across the ministack (*Figure 3D-E*, *Figure 3—figure supplement 1*). When passing through the medial and *trans*-Golgi regions at $t=10$ and 20 min after biotin chase, SBP-GFP-collagenX appeared as double puncta, supporting its cisternal rim localization (*Figure 3D*). 40 min after biotin chase, numerous SBP-GFP-collagenX-positive carriers were observed surrounding the *trans*-Golgi, indicating a process of Golgi exiting (*Figure 3D*). In summary, our findings demonstrated that cisternal rim partitioned large-sized secretory cargos might follow intra-Golgi transport kinetics similar to those of cisternal interior partitioned conventional secretory cargos.

## Distinct intra-Golgi transport velocities for different cargos at the same cisternae

From *Equations 1 and 2*, we derive the following relationship (*Equation 3*):

$$\frac{dLQ}{dt} = \frac{ln2}{t_{intra}}\left(y_0 - LQ\right) \tag{3}$$

**Table 1.** Intra-Golgi transport kinetics of secretory RUSH reporters in Golgi ministacks.

See the legend of *Figure 2* for details. #1–4 indicate independent replicates. All data were acquired from HeLa cells except for those labeled with '293T,' which were acquired using 293T cells. Except for TNFα-SBP-GFP and SBP-GFP, which use Ii-streptavidin as the ER hook, all RUSH reporters employ signal sequence fused streptavidin-KDEL as their ER hook. Superscripts 1 and 2 indicate data were re-analyzed from previous publications, (*Tie et al., 2016*) and (*Sun et al., 2020*), respectively. WT, wild type. $t_{intra}$ (intra-Golgi transport time), $y_0$, and adj. $R^2$ was calculated by fitting measured $LQ$ vs. time kinetics data to *Equation 1*. $dLQ/dt$ (at $LQ = 0.40$) was calculated by *Equation 3*, and converted to nm/min by multiplying 274 nm per $LQ$ unit. *SD*, standard deviation.

| RUSH reporter | $t_{intra}$ (min) Mean ±SD | $t_{intra}$ (min) | $y_0$ | Adj. $R^2$ | $dLQ/dt$ (at LQ = 0.40) | $dLQ/dt$ (at LQ = 0.40) (nm/min) |
|---|---|---|---|---|---|---|
| SBP-GFP-CD59 #1 (293T) | | 0.4 | 0.66 | 0.94 | 0.244 | 66.7 |
| SBP-GFP-CD59 #2 (293T) | 5±4 | 7.3 | 0.63 | 0.92 | 0.022 | 6.0 |
| SBP-GFP-CD59 #3 (293T) | | 6.8 | 0.79 | 0.85 | 0.040 | 10.9 |
| TNFα-SBP-GFP[1] | 6.4 | 6.4 | 1.09 | 0.94 | 0.075 | 20.5 |
| TfR-SBP-GFP | 7 | 7.0 | 1.16 | 0.99 | 0.075 | 20.6 |
| SBP-GFP-CD8a-furin-Y+AC #1[1] | | 4.9 | 0.83 | 0.99 | 0.061 | 16.7 |
| SBP-GFP-CD8a-furin-Y+AC #2[1] | 7±1 | 7.1 | 1.00 | 0.99 | 0.059 | 16.0 |
| SBP-GFP-CD8a-furin-Y+AC #3[1] | | 7.7 | 1.04 | 0.92 | 0.058 | 15.8 |
| SBP-GFP-Tac #1[2] | 7±2 | 6.0 | 0.86 | 0.98 | 0.053 | 14.6 |
| SBP-GFP-Tac #2 | | 8.2 | 0.91 | 0.98 | 0.043 | 11.8 |
| SBP-GFP-E-cadherin #1[1] | | 7.6 | 0.98 | 0.99 | 0.053 | 14.5 |
| SBP-GFP-E-cadherin #2[1] | 8±1 | 8.7 | 1.15 | 0.96 | 0.060 | 16.5 |
| SBP-GFP-E-cadherin #3[1] | | 6.5 | 1.11 | 0.98 | 0.075 | 20.6 |
| SBP-GFP-collagenX #1 | | 9.0 | 1.00 | 0.94 | 0.046 | 12.7 |
| SBP-GFP-collagenX #2 | 8±1 | 9.5 | 0.84 | 0.97 | 0.032 | 8.8 |
| SBP-GFP-collagenX #3 | | 6.9 | 0.91 | 0.97 | 0.051 | 14.0 |
| SBP-GFP | 10.6 | 10.6 | 0.74 | 0.98 | 0.022 | 6.1 |
| SBP-GFP-CD8a-furin-YA #1[1] | | 5.9 | 1.49 | 0.98 | 0.13 | 35.0 |
| SBP-GFP-CD8a-furin-YA #2[1] | 11±5 | 11.1 | 1.51 | 0.93 | 0.069 | 19.0 |
| SBP-GFP-CD8a-furin-YA #3[1] | | 16.3 | 1.48 | 0.95 | 0.046 | 12.6 |
| SBP-GFP-CD59[1] | 13.0 | 13.0 | 1.03 | 0.97 | 0.034 | 9.2 |
| SBP-GFP-Tac-TC #1 (293T) | | 14.7 | 0.95 | 0.97 | 0.026 | 7.1 |
| SBP-GFP-Tac-TC #2 (293T) | 14.2±0.6 | 14.3 | 1.00 | 0.98 | 0.029 | 8.0 |
| SBP-GFP-Tac-TC #3 (293T) | | 13.6 | 0.98 | 0.89 | 0.030 | 8.1 |
| SBP-GFP-Tac-TC #1[2] | 17±3 | 15.5 | 1.05 | 0.97 | 0.029 | 8.0 |
| SBP-GFP-Tac-TC #2 | | 19.1 | 1.16 | 0.94 | 0.028 | 7.6 |
| SBP-GFP-CD8a-furin-WT #1[1] | | 23.6 | 1.63 | 0.96 | 0.036 | 9.9 |
| SBP-GFP-CD8a-furin-WT #2[1] | 18±5 | 18.7 | 1.58 | 0.97 | 0.044 | 12.0 |
| SBP-GFP-CD8a-furin-WT #3[1] | | 13.1 | 1.55 | 0.97 | 0.061 | 16.7 |
| SBP-GFP-CD8a-furin-AC #1[1] | | 17.9 | 1.73 | 0.99 | 0.052 | 14.1 |
| SBP-GFP-CD8a-furin-AC #2[1] | 21±5 | 26.9 | 1.50 | 1.00 | 0.028 | 7.8 |
| SBP-GFP-CD8a-furin-AC #3[1] | | 23.2 | 1.58 | 0.96 | 0.035 | 9.7 |
| SBP-GFP-CD8a-furin-AC #4[1] | | 16.8 | 1.62 | 0.99 | 0.050 | 13.8 |

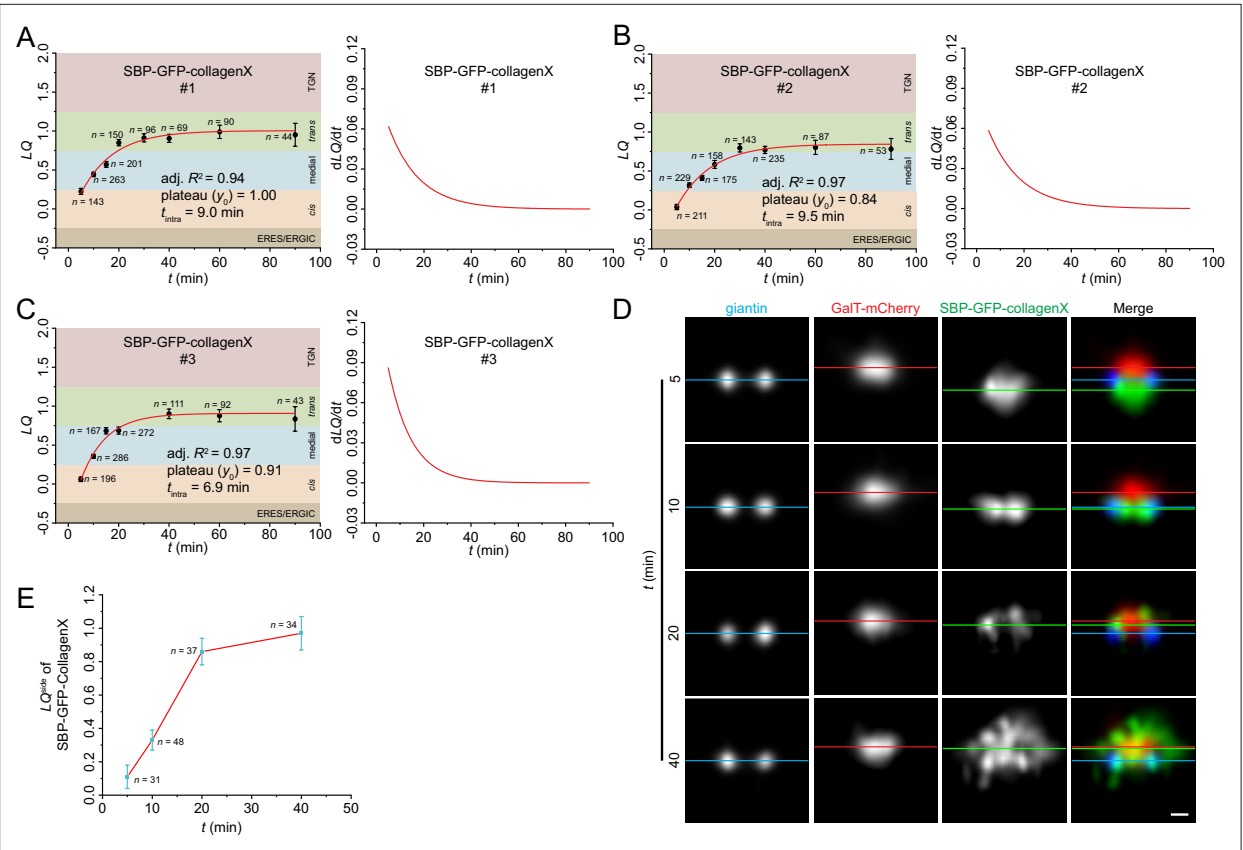

**Figure 3.** The intra-Golgi transport of SBP-GFP-collagenX was quantified by Golgi localization by imaging centers of mass (GLIM) and visualized by side averaging. (**A–C**) GLIM. HeLa cells transiently co-expressing the RUSH reporter, SBP-GFP-collagenX, and GalT-mCherry were imaged during biotin chase (**t**) and subjected to GLIM. See *Figure 2* legend for details. (**D**) Side averaging. HeLa cells transiently co-expressing the RUSH reporter, SBP-GFP-collagenX, and GalT-mCherry were incubated with nocodazole for 3 hr. This was followed by a chase with nocodazole, cycloheximide, and biotin for variable time durations (**t**) before fixation and giantin immunostaining. Images were acquired using Airyscan microscopy and subjected to side averaging guided by giantin double puncta. Blue, red, and green horizontal lines represent the center of mass positions of giantin, GalT-mCherry, and SBP-GFP-collagenX, respectively. Scale bar, 200 nm. (**E**) The $LQ^{side}$ vs. biotin chase time (**t**) plot shows the transition of SBP-GFP-collagenX from the *cis* to the *trans*-region of the Golgi ministack. $LQ^{side}$ is an approximate metric corresponding to $LQ$ (see Materials and methods). *n*, the number of ministacks quantified. Error bar, SEM.

The online version of this article includes the following source data and figure supplement(s) for figure 3:

**Source data 1.** *LQ* data employed in plots presented in *Figure 3*.

**Figure supplement 1.** Example images used for side averaging in *Figure 3D*.

In our previous work, through side-averaging, we determined that one *LQ* unit corresponds to 274 nm (*Tie et al., 2022*). Hence, we scaled *Equation 3* by 274 nm to derive the instantaneous intra-Golgi transport velocity in nm/min. When plotting the instantaneous intra-Golgi transport velocity (nm/min) against the *LQ* for selected RUSH reporters, as seen in *Figure 4A*, we observed that different secretory cargos exhibit varied transport velocities even within the same cisternae or at the same *LQ* values. At LQ = 0.40, corresponding to the medial-Golgi region, our RUSH reporters' instantaneous intra-Golgi transport velocities are calculated in *Table 1* according to *Equation 3*. For instance, instantaneous intra-Golgi transport velocities of SBP-GFP, SBP-GFP-Tac-TC, SBP-GFP-CD59, SBP-GFP-Tac, and TNFα-SBP-GFP are 6.1, 8.0, 9.2, 14.6, and 20.5 nm/min, respectively (*Table 1*). These findings highlighted that different secretory cargos possess distinct intra-Golgi transport velocities within the same Golgi cisternae, challenging the prediction made by the classic cisternal progression model.

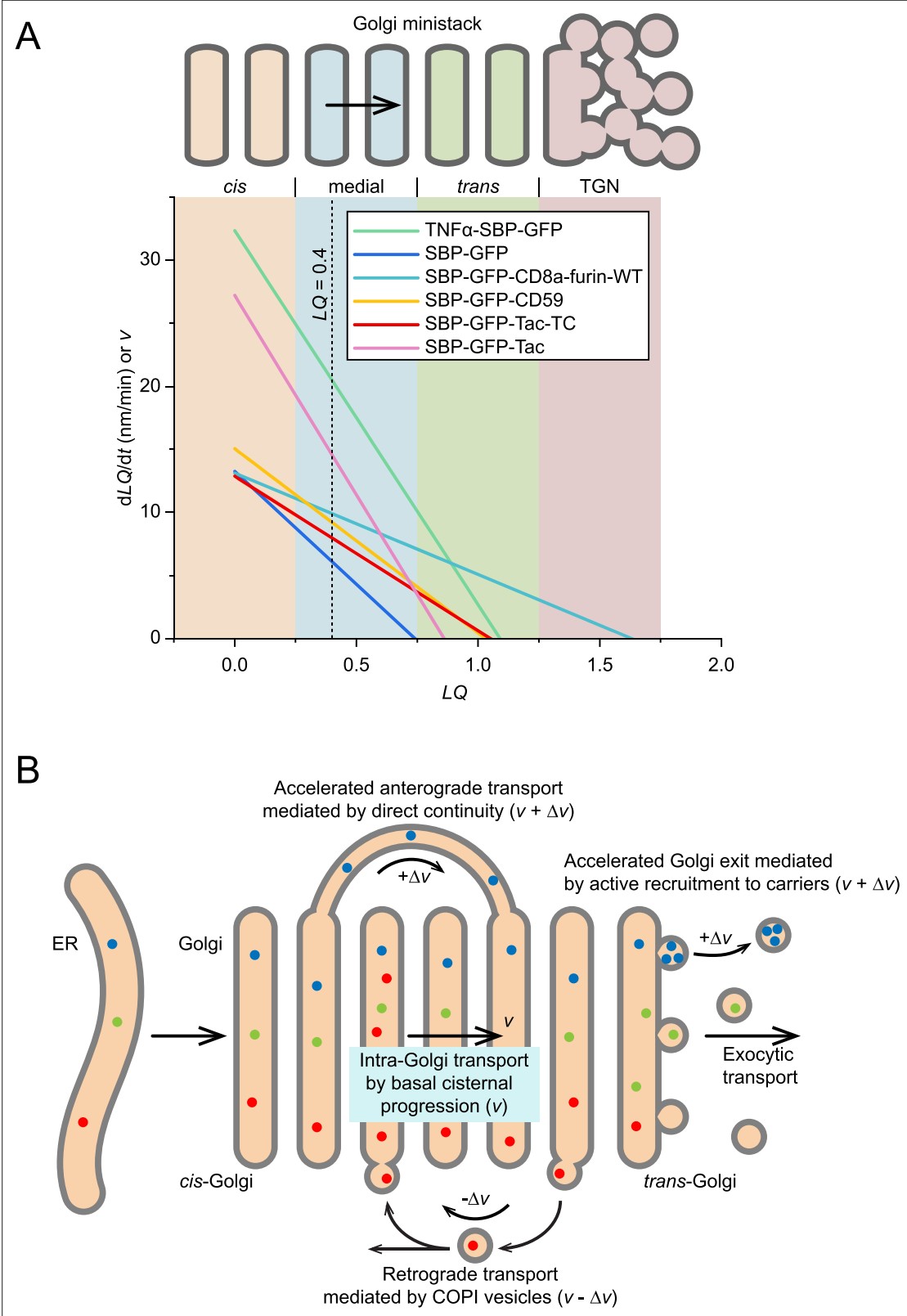

**Figure 4.** Different secretory cargos exhibit distinct intra-Golgi transport velocities within the same cisternae of Golgi ministacks. (**A**) Plots of d$LQ$/d$t$ vs. $LQ$ for selected RUSH reporters reveal diverse intra-Golgi transport velocities at the same cisternae. These plots were constructed based on **Equation 3**, using parameters from **Table 1**. Different regions within the Golgi — *cis* (−0.25≤$LQ$<0.25), medial (0.25≤$LQ$<0.75), *trans*-Golgi (0.75≤$LQ$<1.25), and TGN (1.25≤$LQ$) — are color-shaded for easier identification. To convert d$LQ$/d$t$ to nm/min, we multiplied d$LQ$/d$t$ by 274 nm per $LQ$ unit. A dotted vertical

*Figure 4 continued*

line marks the *LQ* value of 0.40. (**B**) Modifying the classic cisternal progression model to explain the distinct intra-Golgi transport velocities at the same cisternae. According to this model, Golgi cisternae progress at a constant velocity (**v**). Direct-continuity-based anterograde transport could accelerate the intra-Golgi transport velocity (*v + Δv*), while the COPI-mediated retrograde transport could reduce it (*v - Δv*).

The online version of this article includes the following figure supplement(s) for figure 4:

**Figure supplement 1.** Acquiring Golgi residence times of SBP-GFP-CD59 and GFP-CD8a-TC.

## Golgi residence times vary significantly among different secretory cargos

Once cargos transit through the Golgi stack, the classic cisternal progression model predicts that cargos depart the Golgi in a linear kinetics. However, studies have demonstrated that cargos exit the Golgi by following the first-order exponential kinetics (*Hirschberg et al., 1998*; *Patterson et al., 2008*; *Sun et al., 2020*). While we could apply a hypothetical rate-limiting step to the classic cisternal progression and stable compartment models to rationalize the exponential kinetics of cargo exit (*Luini, 2011*), the classic cisternal progression model encounters more significant challenges. First, the first-order exponential Golgi exit implies that clearing a synchronized wave of secretory cargo from the *trans*-cisternae would take an indefinite time, which is inconsistent with the transient nature of the *trans*-cisternae as described by the classic cisternal progression model. Second, since cargos are considered passive in the classic cisternal progression model, the exit kinetics, as measured by the Golgi residence times, should be the same across all secretory cargos.

The Golgi residence time is a cargo's duration at the *trans*-Golgi cisternae before exit and is a metric for Golgi retention (*Sun et al., 2021*; *Sun et al., 2020*). The first step to determine this metric involves synchronizing a transmembrane protein at the Golgi. For Golgi transmembrane resident proteins like glycosyltransferases and secretory transmembrane proteins with a substantial Golgi localization at the steady state, such as GFP-Tac-TC, synchronization is not required. However, for secretory transmembrane cargos lacking a significant Golgi pool at the steady state, such as TfR-GFP, GFP-Tac, and RUSH reporters like TNFα-SBP-GFP, synchronization is achieved through a 20 °C temperature block. Subsequently, live imaging at 37 °C is performed to capture the Golgi fluorescence intensity decay of the protein in the presence of cycloheximide. The Golgi residence time is calculated as the half-time ($t_{1/2}$) by fitting the intensity decay to a first-order exponential function.

Our extensive measurement demonstrated that Golgi residence times of secretory cargos display a wide range of values (*Table 2*). For example, TNFα-SBP-GFP has a Golgi residence time of 6.0±0.4 min (mean ± SEM, n=73), one of the shortest, while SBP-GFP-CD59 has a Golgi residence time of 16±2 min

**Table 2.** Golgi residence times of transmembrane secretory cargos and Golgi glycosyltransferases in native Golgi (without nocodazole).

Superscripts 1 and 2 indicate data were from previous publications, (*Sun et al., 2020*) and (*Sun et al., 2021*), respectively. The two RUSH reporters, SBP-GFP-CD59 and TNFα-SBP-GFP, employ signal sequence fused streptavidin-KDEL and Ii-streptavidin as the ER hook, respectively. *n*, the number of quantified cells; SEM, standard error of the mean.

| Transmembrane protein | Golgi residence time ($t_{1/2}$) | *n* | SEM |
| --- | --- | --- | --- |
| TNFα-SBP-GFP[1] | 6.0 min | 73 | 0.4 min |
| GFP-CD8a[1] | 7.8 min | 26 | 0.7 min |
| TfR-GFP[1] | 10 min | 12 | 1 min |
| GFP-Tac[1] | 16 min | 45 | 2 min |
| SBP-GFP-CD59 | 16 min | 29 | 2 min |
| GFP-Tac (5 A)[1] | 47 min | 57 | 3 min |
| GFP-CD8a-TC | 66 min | 19 | 7 min |
| GFP-Tac-TC[1] | 3.4 h | 22 | 0.3 h |
| MGAT2-GFP[2] | 4.9 h | 26 | 0.5 h |
| ST6GAL1-GFP[1] | 5.3 h | 21 | 0.6 h |

(mean ± SEM, n=29) (*Figure 4—figure supplement 1*). Hence, our data suggest that different secretory cargos reside at the *trans*-Golgi cisternae for varying durations, contradicting predictions from the classic cisternal progression model.

## Cargos exhibiting prolonged Golgi residence times suggest the *trans*-Golgi interior might be a stable domain

The classic cisternal progression model could be modified to explain the diverse intra-Golgi transport kinetics. For example, accelerated anterograde or retrograde transport might be introduced on top of the basal cisternal progression to account for the wide range of intra-Golgi transport velocities and Golgi residence times we observed (*Figure 4B*). Accelerated anterograde transport mechanisms might include continuity-based direct diffusion across cisternae via heterologous cisternal connections (*Beznoussenko et al., 2014*; *Marsh et al., 2004*; *Trucco et al., 2004*). Hence, secretory cargo with such a mechanism would have a faster intra-Golgi transport velocity. This mechanism might explain the rapid and diverse intra-Golgi transport velocity of secretory cargos such as VSVG, insulin, albumin, and alpha1-antitrypsin (*Beznoussenko et al., 2014*; *Marsh et al., 2004*; *Trucco et al., 2004*). Similarly, active recruitment to exocytic carriers budding at the *trans*-Golgi possibly might shorten the Golgi residence time. On the other hand, COPI-coated carriers might facilitate retrograde intra-Golgi transport to counter the cisternal progression, accounting for the slow intra-Golgi transport velocity and prolonged Golgi residence time of cargos. Indeed, COPI has been known to maintain certain glycosyltransferases' Golgi retention by direct or indirect interactions (*Ali et al., 2012*; *Eckert et al., 2014*; *Liu et al., 2018*; *Pereira et al., 2014*; *Rizzo et al., 2021*; *Schmitz et al., 2008*; *Tu et al., 2008*). However, such a retrograde mechanism requires the interaction between secretory cargo and the COPI coat.

Earlier, we identified truncation mutants of two typical secretory cargos, GFP-Tac (*Sun et al., 2020*) and GFP-CD8a, which reside in the Golgi nearly as stably as Golgi glycosyltransferases. GFP-Tac and GFP-CD8a are type I transmembrane proteins comprising from their N- to C-termini a signal sequence, GFP, luminal domain, transmembrane domain (TMD), and cytosolic tail (*Figure 5A*). Previously, we and others demonstrated that a fully glycosylated luminal domain can function as a Golgi exit signal (*Gut et al., 1998*; *Sun et al., 2020*). Their Golgi residence time was documented at 16±2 min (mean ± SEM, n=45) and 7.8±0.7 min (mean ± SEM, n=26), respectively (*Table 2*). However, after truncating the luminal domain, the Golgi residence time for the resultant chimeras, GFP-Tac-TC and GFP-CD8a-TC, extended significantly to 3.4±0.3 hr (mean ± SEM, n=22) and 66±7 min (mean ± SEM, n=19) (*Table 2*; *Figure 4—figure supplement 1*), respectively. Consequently, their long Golgi residence times ensure a significant steady-state Golgi pool.

The following evidence suggests that GFP-Tac-TC and GFP-CD8a-TC behave like bona fide Golgi resident transmembrane proteins. First, their Golgi residence times, 3.4 hr and 66 min, are comparable to that of a typical Golgi glycosyltransferase, such as ST6GAL1 (5.3±0.6 hr, mean ± SEM, n=21) (*Sun et al., 2021*). Second, their *LQ*s, 0.94±0.02 (mean ± SEM, n=118) and 0.86±0.05 (mean ± SEM, n=76), combined with their cisternal interior localization (*Figure 5B*; *Sun et al., 2020*), indicate that they primarily reside within the interior of the *trans*-cisternae, similar to many Golgi glycosyltransferases (*Tie et al., 2018*). Additionally, GFP-Tac-TC follows GalT-mCherry to localize to the ER under the BFA treatment reversibly (*Sun et al., 2020*).

According to the modified cisternal progression model above, GFP-Tac-TC could have an active retrieval mechanism mediated by an unidentified signal in its cytosolic tail or TMD to counter the cisternal progression. Considering that GFP-Tac-TC's *LQ* (0.94) is close to its Golgi exit site, $y_0$ (0.95–1.16) (*Table 1*; *Figure 2*, *Figure 2—figure supplement 1M-O*), measured by its RUSH version, there are two possible retrieval pathways, acting at either post-Golgi or intra-Golgi stages. To test if GFP-Tac-TC possesses a Golgi retrieval pathway after its Golgi exit, we incubated HeLa cells transiently expressing MGAT1-GFP (negative control) (*Sun et al., 2021*), MGAT2-GFP (positive control) (*Sun et al., 2021*), GFP-Tac, or GFP-Tac-TC with a recombinant mCherry-fused anti-GFP nanobody (VHH-anti-GFP-mCherry) continuously for 8 hr (*Figure 5C*). By binding to the cell surface-exposed GFP, VHH-anti-GFP-mCherry serves as a sensitive probe to track the endocytic trafficking itinerary of the above GFP-fused transmembrane proteins.

We found that VHH-anti-GFP-mCherry localized to the Golgi in a significant fraction of MGAT2-GFP-expressing cells (32%, n=266) but not in MGAT1-GFP cells (0%, n=102) after 8 hr of continuous internalization (*Figure 5C*). This result is consistent with our previous report that MGAT2, but not

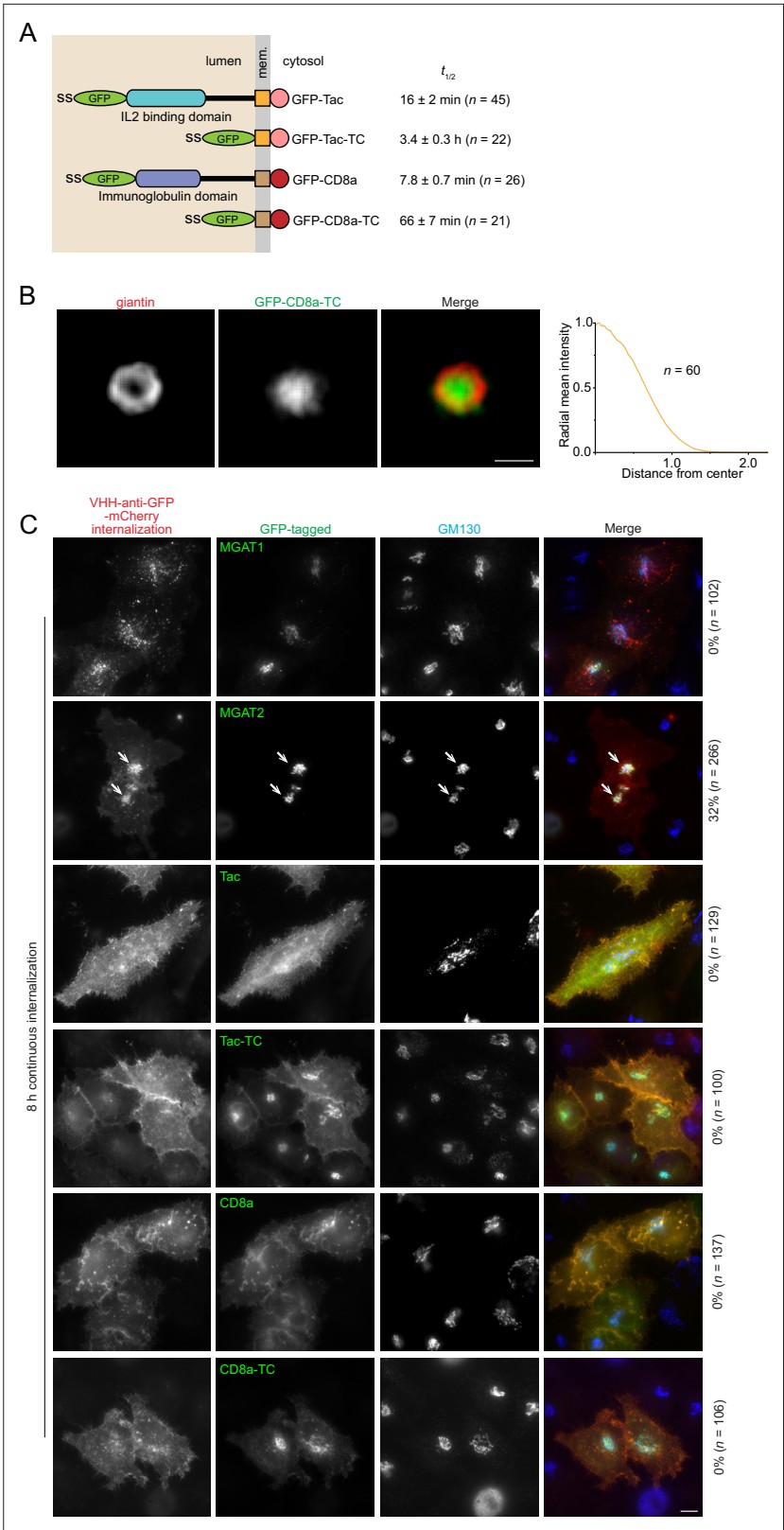

**Figure 5.** GFP-Tac-TC and GFP-CD8a-TC might not have a Golgi retrieval mechanism once exiting the Golgi. (**A**) Domain organization of GFP-tagged Tac, Tac-TC, CD8a, and CD8a-TC. ss, signal sequence. Mem., membrane. $t_{1/2}$ values are from **Table 2** and shown as mean ± SEM. *n*, the number of cells analyzed. (**B**) GFP-CD8a-TC localizes to the Golgi cisternal interior at the steady state. HeLa cells transiently expressing GFP-CD8a-TC were treated

*Figure 5 continued on next page*

*Figure 5 continued*

with nocodazole to induce the formation of ministacks before fixation and immunostaining for the endogenous giantin. En face averaging images of giantin and GFP-CD8a-TC are shown on the left. Scale bar, 500 nm. The radial mean intensity profile of en face averaged GFP-CD8a-TC is shown on the right. The x-axis represents the distance from the center of fluorescence mass (normalized to the giantin radius), and the y-axis represents the radial mean intensity (normalized). *n*, the number of averaged ministacks. (**C**) The plasma membrane-localized GFP-Tac-TC and GFP-CD8a-TC are not retrieved to the Golgi. HeLa cells transiently expressing the indicated GFP-tagged protein were incubated continuously with VHH-anti-GFP-mCherry for 8 hr before fixation and immunostaining for the endogenous Golgi marker, GM130. Images were acquired under a wide-field microscope. The internalized VHH-anti-GFP-mCherry has negative Golgi localization for all GFP-tagged proteins except for MGAT2 (arrows), a positive control. The percentage of cells showing the Golgi localization of VHH-anti-GFP-mCherry is labeled on the right. *n*, the number of cells analyzed. Scale bar, 10 μm.

MGAT1, has a Golgi retrieval mechanism (*Sun et al., 2021*), although its molecular mechanism is still unknown. In contrast, VHH-anti-GFP-mCherry did not localize to the Golgi in cells expressing other constructs, including GFP-Tac-TC, suggesting that GFP-Tac-TC might not possess a post-Golgi retrieval mechanism targeting the Golgi (*Figure 5C*). Next, we reason that given GFP-Tac's short Golgi residence time (16 min), its cytosolic tail and TMD might not facilitate any active intra-Golgi retrograde transport mechanism, such as COPI coat binding, to recycle GFP-Tac-TC from Golgi exiting. Since GFP-Tac and GFP-Tac-TC share identical TMD and cytosolic tail sequences, if such a retrograde mechanism existed, GFP-Tac would have a similar Golgi residence time to GFP-Tac-TC. The same observation and reasoning also apply to GFP-CD8a-TC (*Figure 5C*).

Therefore, we argue that GFP-Tac-TC and GFP-CD8a-TC might not have retrieval signals to facilitate their Golgi residence, although proving a protein does not possess a transport signal is challenging. Since the cisternal interior is continuous with the cisternal rim both in membrane and lumen, our findings suggest that the cisternal interior at the *trans*-Golgi might be a stable domain. In summary, our data implies that retention within the *trans*-side stable domain, rather than continuous retrieval to counter the cisternal progression (treadmilling), could be the primary mechanism for the long Golgi residence times of GFP-Tac-TC and GFP-CD8a-TC.

## The Golgi maintains its stacked organization after 30 min BFA treatment

COPI functions in the retrograde direction at the ER-Golgi interface and within the Golgi (*Glick and Luini, 2011*; *Popoff et al., 2011*; *Rabouille and Klumperman, 2005*). According to the classic cisternal progression model, COPI-mediated retrograde intra-Golgi transport recycles resident transmembrane proteins, such as glycosyltransferases and transport machinery components. This model predicts that upon the compromise of COPI, intra-Golgi recycling would stop, and the Golgi stack would continuously lose its materials and eventually disappear, depending on the cisternal progression rate. In addition to its retrograde role, COPI has also been documented to function in the anterograde ER-to-Golgi transport (*Monetta et al., 2007*; *Weigel et al., 2021*).

To test the role of COPI in the Golgi organization, we employed BFA, a small molecule fungal metabolite that rapidly dissociates COPI and clathrin from the Golgi, inhibits the ER-to-Golgi trafficking, and causes the fusion of the Golgi with the ER in 10–20 min (*Klausner et al., 1992*). It does so by rapidly inactivating class I ARFs, which recruit COPI and clathrin coats to the Golgi membrane (*Donaldson et al., 1992*; *Helms and Rothman, 1992*). In contrast to the native Golgi, nocodazole-induced Golgi ministacks have been documented to be more resistant to BFA, and, hence, the disappearance of Golgi occurs at a much later time (>30 min) (*Lippincott-Schwartz et al., 1990*). However, it is unclear if the nocodazole-induced Golgi maintains a similar stacked organization under the BFA treatment.

To address this, we studied the *LQ*s of several Golgi markers following 30–60 min of BFA treatment (*Figure 6A-E*, *Figure 6—figure supplement 1A and B*). The fast dissociation of Arf1-GFP from Golgi ministacks confirmed the effectiveness of BFA (*Figure 6—figure supplement 1A*). We also observed that extended BFA treatment considerably reduced the number of intact Golgi ministacks. We measured the *LQ*s of six transmembrane Golgi markers, GFP-golgin-84, GS15, ST6GAL1-GFP, TGN46, CD8a-furin, and CD8a-CI-M6PR (*Figure 6A-F*, *Figure 6—figure supplement 1B*). GS15's

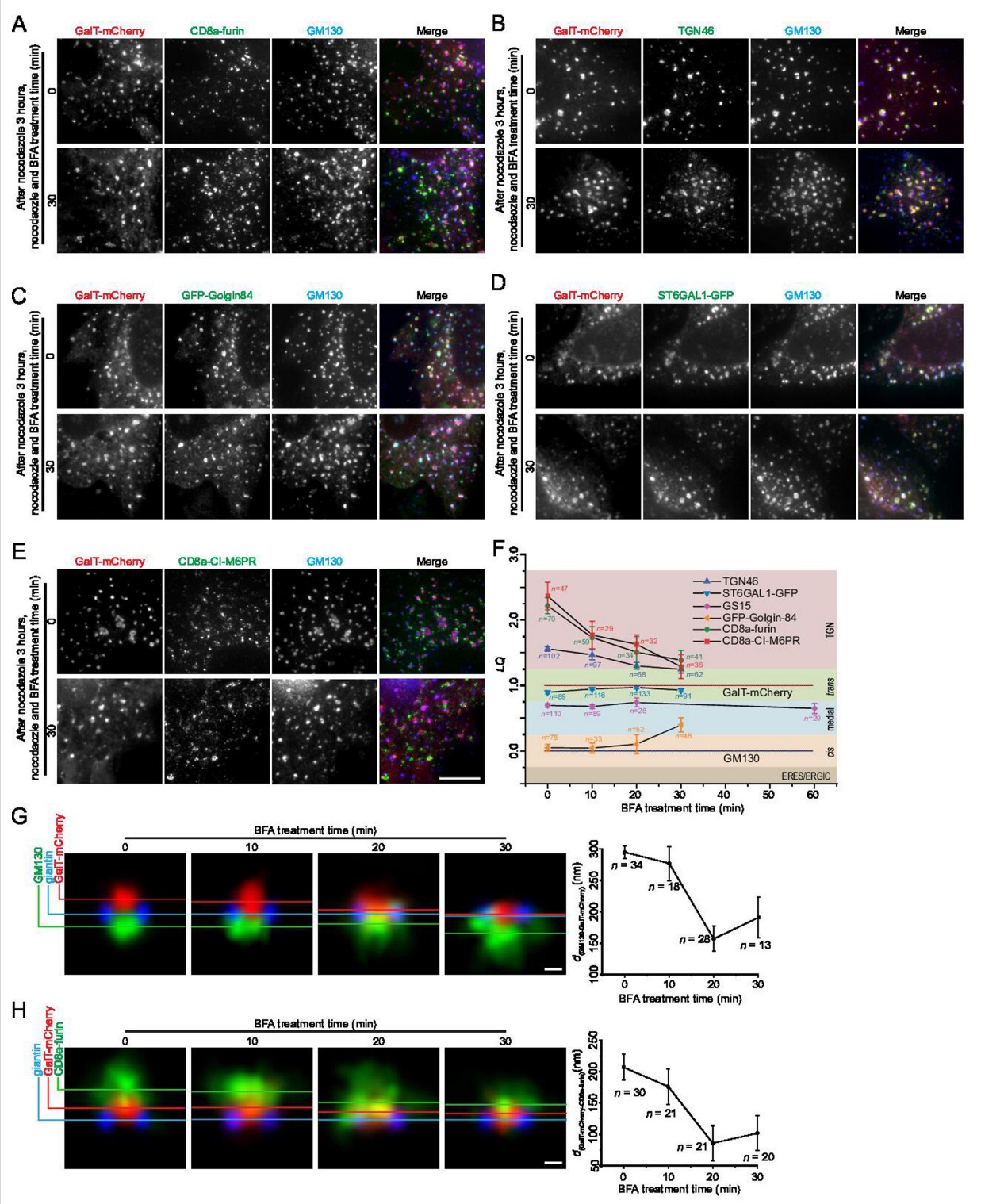

**Figure 6.** The organization of Golgi ministacks under the brefeldin A (BFA) treatment. (**A–E**) HeLa cells transiently co-expressing GalT-mCherry and indicated GFP or CD8a-tagged Golgi proteins were incubated with nocodazole for 3 hr. This was followed by additional treatment with nocodazole and 5 μM BFA for variable durations before fixation. Endogenous TGN46 and GM130 was immunostained. Images were acquired by wide-field microscopy. Representative images at 0 and 30 min are shown. Scale bar, 10 μm. (**F**) *LQ*s of different Golgi proteins were determined using Golgi localization by imaging centers of mass (GLIM). The dotted lines represent *LQ*s for GM130 (LQ = 0.00) and GalT-mCherry (LQ = 1.00). *n*, the number of analyzed Golgi ministacks. (**G–H**) HeLa cells transiently expressing GalT-mCherry alone (**G**) or together with CD8a-furin (**H**) were incubated with nocodazole for 3 hr.

*Figure 6 continued on next page*

*Figure 6 continued*

This was followed by additional treatment with nocodazole and 5 μM BFA for variable durations before immunostaining endogenous giantin or GM130. After Airyscan imaging, images were subjected to side averaging. Horizontal color lines indicate centers of mass of corresponding side-averaged Golgi proteins. $d_{(GM130\text{-}GalT\text{-}mCherry)}$ and $d_{(GalT\text{-}mCherry\text{-}CD8a\text{-}furin)}$ in nm are plotted against the BFA treatment time in the right panels. *n*, the number of ministacks quantified. Error bar, SEM. Scale bar, 200 nm.

The online version of this article includes the following source data, source code, and figure supplement(s) for figure 6:

**Source code 1.** Translation_Rotation_Gaussian_Fitting.

**Source data 1.** *LQ*, $d_{(GM130\text{-}GalT\text{-}mCherry)}$, and $d_{(GalT\text{-}mCherry\text{-}CD8a\text{-}furin)}$ data employed in plots presented in *Figure 6*.

**Figure supplement 1.** The effect of brefeldin A (BFA) treatment on Golgi markers.

*LQ* was monitored for 60 min, with the rest for 30 min. Strikingly, we observed that Golgi puncta are still stacked as all measured *LQ*s exhibited a relative arrangement similar to those of the pre-BFA treatment (*t*=0 min) (*Figure 6F*). For example, *LQ*s of ST6GAL1 and GS15 remained largely unchanged after 30 min BFA treatment. Although the *LQ* of golgin-84 increased from 0.2 to 0.4, it still remained between the *LQ*s of GM130 and GS15. The stacked organization of BFA-treated ministacks was further confirmed by the side-averaging images of GM130, Giantin, GalT-mCherry, and CD8a-furin (*Figure 6G and H*).

However, BFA treatment altered the physical dimensions of ministacks. We defined the distance between two Golgi markers as the distance between their Gaussian peak centers in the axial line intensity profile (see the Materials and methods). We observed the distance from GM130 to GalT-mCherry, $d_{(GM130\text{-}GalT\text{-}mCherry)}$, decreased from 300±10 nm (mean ± SEM, n=34)–190±30 (mean ± SEM, n=13) after 30 min of BFA treatment, indicating axial shrinkage of ministacks (*Figure 6G*). Furthermore, *LQ*s of all three TGN markers, TGN46, CD8a-furin, and CD8a-CI-M6PR, decreased and approached that of the *trans*-Golgi, suggesting a collapse of the TGN (*Figure 6F*), possibly due to the dissociation of clathrin and its adaptor proteins. This collapse of the TGN was further supported by the side-averaging images, in which CD8a-furin gradually approached the *trans*-Golgi labeled by GalT-mCherry over time (*Figure 6H*). We found the distance from GalT-mCherry to CD8a-furin, $d_{(GalT\text{-}mCherry\text{-}CD8a\text{-}furin)}$, decreased from 200±20 nm (mean ± SEM, n=30)–100±30 nm (mean ± SEM, n=20) after 30 min of BFA treatment.

Despite dramatic changes in the TGN and shrinkage of the axial length of the Golgi minitack, our data demonstrate that the Golgi maintains its stacked organization for at least 30–60 min, even in the absence of COPI-mediated intra-Golgi retrograde transport. Our findings suggest that the Golgi might not be a dynamic equilibrium between the cisternal progression and retrograde trafficking and argue for a by-default stable nature of the Golgi stack. It is worth noting that the cellular effect of BFA is complex and pleiotropic. For example, in addition to Arfs, it can inhibit lipid metabolic enzymes (*De Matteis et al., 1994*). So, the dissociation of COPI might not be the sole factor responsible for our observations.

## Discussion

### Our data supports the rim progression model, a modified version of the stable compartment model

Previously, we discovered that Golgi glycosyltransferases tend to localize to the cisternal interior, while trafficking machinery components localize primarily to the cisternal rim (*Tie et al., 2018*). Combined with our current observations of differential intra-Golgi transport velocities among distinct secretory cargos (*Figures 2–4*, *Figure 2—figure supplement 1*) and the stable *trans*-Golgi interior localization of GFP-Tac-TC (*Sun et al., 2020*) and GFP-CD8a-TC (*Figure 5B* and *Table 2*), these findings support the modified version of the stable compartment model – the rim progression model (*Lavieu et al., 2013*; *Pfeffer, 2010*; *Volchuk et al., 2000*). This model explains the retention of Golgi glycosyltransferases in the cisternal interior, given that many do not recycle via COPI-coated vesicles (*Liu et al., 2018*) or possess a post-Golgi retrieval pathway (*Sun et al., 2021*). It also readily accounts for the diverse intra-Golgi transport velocities of secretory cargos by suggesting that each cargo type may have a different retention time within the stable cisternal interior domain.

The Golgi glycosyltransferases at the cisternal interior might assemble a dense protein matrix based on fluorescence microscopy (*Tie et al., 2018*) and EM data (*Engel et al., 2015*). It is tempting

to speculate that the enzyme matrix could provide the molecular basis for the stable cisternal interior domain and functionally mirror a gel-filtration chromatography matrix with a defined porosity. Hence, large secretory cargos, such as FM4 aggregates and collagenX, are excluded from the interior, where the enzyme matrix localizes, while small secretory cargos can enter and become kinetically trapped there. The progressive reduction in intra-Golgi transport of secretory cargo might result from the enzyme matrix's retention at the *trans*-Golgi. As the secretory cargos progress along the Golgi stack from the *cis* to the *trans*-side, more and more cargos become temporarily retained in the *trans*-Golgi region, gradually reducing their overall intra-Golgi transport velocity. If the release or Golgi exit of these cargos from the enzyme matrix follows a constant probability per unit time, i.e., a first-order kinetics process, the rate of cargo exiting from the Golgi should follow the first-order exponential function. Since the mechanism underlying intra-Golgi transport kinetics reflects fundamental molecular and cellular processes of the Golgi, further experimental data are essential to rigorously test this hypothesis.

## Limitations of the study

We introduced new quantitative data on the intra-Golgi transport dynamics. However, our study has limitations. First, our approach relied on the overexpression of fluorescence protein-tagged cargos. The synchronized release of a large amount of cargo could significantly saturate and skew the intra-Golgi transport. Second, we utilized nocodazole-induced ministacks instead of the native Golgi to analyze the intra-Golgi transport, which could raise concerns about the impact of depolymerizing microtubules on the intra-Golgi transport and Golgi organization. Third, with the exception of furin and its mutants, all RUSH reporters used in this study are constitutive secretory cargos. As a result, the intra-Golgi transport dynamics observed here might not reflect those of regulated secretion, which involves the synchronized release of a large quantity of cargo in response to a specific signal.

Our findings suggest that the Golgi cisternal interior might be a stable domain, therefore, supporting a modified version of the stable compartment model, the rim progression model. However, we do not think our data alone can resolve the two models, which have been the subject of debate for several decades. Further refinement of the classic cisternal progression model might also account for our data. We anticipate that further development using our approach will provide more systematic data to test and refine future intra-Golgi transport models. Moreover, we hope that our study will stimulate further research into this longstanding and intriguing question.

# Materials and methods

## Key resources table

| Reagent type (species) or resource | Designation | Source or reference | Identifiers | Additional information |
|---|---|---|---|---|
| Cell line (*Homo sapiens*) | HeLa cell | ATCC | ATCC: CCL-2; RRID:CVCL_0030 | |
| Cell line (*Homo sapiens*) | HEK 293T cell | ATCC | ATCC: CRL-3216 RRID:CVCL_0063 | |
| Antibody | Anti-GM130 C-terminus (mouse monoclonal) | BD Bioscience | Cat#: 610822; RRID:AB_398141 | IF (1:500) |
| Antibody | Anti-giantin C-terminus (rabbit, polyclonal) | BioLegend | Cat#: 924302; RRID:AB_2565451 | IF (1:1000) |
| Antibody | Anti-CD8a (mouse, monoclonal) | Thermo Fisher Scientific | Cat#:14-0086-80; RRID:AB_467092 | IF (1:500) |
| Antibody | Anti-TGN46 (rabbit, polyclonal) | Abcam | Cat#: ab50595; RRID:AB_2203289 | IF (1:200) |
| Antibody | Anti-GS15 (mouse, monoclonal) | BD Bioscience | Cat#: 610960; RRID:AB_398273 | IF (1:250) |
| Antibody | Anti-myc (mouse, monoclonal) | Santa Cruz Biotechnology | Cat#: sc-40; RRID:AB_627268 | IF (1:200) |

*Continued on next page*

*Continued*

| Reagent type (species) or resource | Designation | Source or reference | Identifiers | Additional information |
|---|---|---|---|---|
| Antibody | Alexa Fluor Plus 680 conjugated donkey anti-mouse IgG (H+L) | Thermo Fisher Scientific | Cat#: A10038; RRID:AB_11180593 | IF (1:500) |
| Antibody | Alexa Fluor 647 conjugated goat anti-rabbit IgG (H+L) | Thermo Fisher Scientific | Cat#: A21244; RRID:AB_2535814 | IF (1:500) |
| Recombinant DNA reagent | Ii-streptavidin_ TNFα-SBP-GFP | PMID:22406856 | RRID:Addgene_65280 | A gift from F. Perez lab (Institut Curie) |
| Recombinant DNA reagent | ss-Strep-KDEL_ss-SBP-GFP-CD8a-Furin | PMID:26764092 | | |
| Recombinant DNA reagent | ss-Strep-KDEL_ss-SBP-GFP-CD8a-Furin-YA | PMID:26764092 | | |
| Recombinant DNA reagent | ss-Strep-KDEL_ss-SBP-GFP-CD8a-Furin-AC | PMID:26764092 | | |
| Recombinant DNA reagent | ss-Strep-KDEL_ss-SBP-GFP-CD8a-Furin-Y+AC | PMID:26764092 | | |
| Recombinant DNA reagent | ss-Strep-KDEL_ss-SBP-GFP-CD59 | PMID:26764092 | RRID:Addgene_222307 | A gift from F. Perez lab (Institut Curie) |
| Recombinant DNA reagent | ss-Strep-KDEL_ TfR-SBP-GFP | PMID:28978644 | | A gift from Bonifacino's lab (NIH) |
| Recombinant DNA reagent | ss-Strep-KDEL_ss-SBP-GFP-E-cadherin | PMID:22406856 | RRID:Addgene_65286 | A gift from F. Perez lab (Institut Curie) |
| Recombinant DNA reagent | ss-Strep-KDEL_ss-SBP-GFP-collagenX | PMID:31142554 | RRID:Addgene_222305 | A gift from F Perez lab (Institut Curie) |
| Recombinant DNA reagent | ss-Strep-KDEL_ss-SBP-GFP-Tac | PMID:32826314 | RRID:Addgene_162505 | |
| Recombinant DNA reagent | ss-Strep-KDEL_ss-SBP-GFP-Tac-TC | PMID:32826314 | RRID:Addgene_162506 | |
| Recombinant DNA reagent | Ii-streptavidin_ ss-SBP-GFP | PMID:22406856 | RRID:Addgene_65277 | A gift from F. Perez lab (Institut Curie) |
| Recombinant DNA reagent | GalT-mCherry | PMID:26764092 | RRID:Addgene_87327 | |
| Recombinant DNA reagent | GFP-Tac | PMID:32826314 | RRID:Addgene_162489 | |
| Recombinant DNA reagent | GFP-Tac-TC | PMID:32826314 | RRID:Addgene_162492 | |
| Recombinant DNA reagent | GFP-CD8a | PMID:32826314 | | GFP-tagged CD8a |
| Recombinant DNA reagent | GFP-CD8a-TC | This paper | | GFP-tagged transmembrane and cytosolic domain of CD8a |
| Recombinant DNA reagent | MGAT1-GFP | PMID:34533190 | RRID:Addgene_163647 | |
| Recombinant DNA reagent | MGAT2-GFP | PMID:34533190 | | |
| Recombinant DNA reagent | CD8a-furin | PMID:24285343 | | |
| Recombinant DNA reagent | GFP-Golgin84 | PMID:12538640 | | A gift from M Lowe lab (University of Manchester) |
| Recombinant DNA reagent | ST6GAL1-GFP | PMID:34533190 | RRID:Addgene_162500 | |
| Recombinant DNA reagent | CD8a-CI-M6PR | PMID:24285343 | | |
| Recombinant DNA reagent | Arf1-GFP | PMID:16890159 | | A gift from FJM van Kuppeveld lab (Utrecht University) |
| Recombinant DNA reagent | ST6GAL1-Dmyc | PMID:30499774 | | |
| Recombinant DNA reagent | Strep-Ii_VSVG-SBP-EGFP | PMID:22406856 | RRID:Addgene_65300 | Addgene plasmid #65300 |
| Chemical compound, drug | Brefeldin A (from *Penicillium brefeldianum*) | Life Technologies Holdings | Cat#:B7450 | 5 µM |
| Chemical compound, drug | Nocodazole | Merck | Cat#:487928 | 33 µM |

*Continued on next page*

*Continued*

| Reagent type (species) or resource | Designation | Source or reference | Identifiers | Additional information |
|---|---|---|---|---|
| Chemical compound, drug | biotin | IBA | Cat#:2-1016-002 | 50 µM |
| Chemical compound, drug | Cycloheximide | Sigma-Aldrich | Cat#:C1988 | 10 µg/mL |
| Software, algorithm | Fiji | NIH | RRID:SCR_002285 | For Image analysis |
| Software, algorithm | Calculation of the LQ | PMID:26764092; PMID:28829416 | | |
| Software, algorithm | Auto-GLIM | This paper | | To calculate LQ automatically (See Materials and Methods) |
| Software, algorithm | Translation_Rotation_Gaussian_Fitting | This paper | | To calculate the axial position Xc of Golgi protein with side view (See *Figure 6—source code 1*) |
| Software, algorithm | Gyradius and intensity normalization | PMID:30499774 | | |
| Software, algorithm | Golgi mini-stack alignment | PMID:30499774 | | |
| Software, algorithm | Radial mean intensity profile | PMID:30499774 | | |
| Software, algorithm | P1-Rotate_Resize_Normaliz | PMID:35467701 | | |
| Software, algorithm | P2-Resize_Add_Line | PMID:35467701 | | |

## DNA plasmids, antibodies, and small molecules

To clone RUSH reporter, SBP-GFP, two PCRs were performed using li-Strep_ss-SBP-EGFP-Ecadherin (a gift plasmid from F. Perez) (*Boncompain et al., 2012*) and pEGFP-C1 (Clontech) as templates and the following primer pairs (Gat gca Ccc ggg agg cgc gcc atg and ctc ctc gcc ctt gct cac acc tgc agg tgg ttc acg) and (Cgt gaa cca cct gca ggt gtg agc aag ggc gag gag and Gat gca tct aga tta ctt gta cag ctc gtc cat), respectively. The mixture of the two purified PCR fragments was used as the template for the third round of PCR amplification using the first and the fourth primer listed above. The resulting PCR fragment was digested by XmaI and XbaI and ligated to XmaI and XbaI digested li-Strep_ss-SBP-EGFP-Ecadherin DNA plasmid. To clone GFP-CD8a-TC, the coding sequence of the TMD and cytosolic tail of CD8a was amplified by PCR using GFP-CD8a (*Sun et al., 2020*) as a template and the following primers, cag tgc ctc gag gac ttc gcc tgt gat atc ta and gac cgt gaa ttc TTA GAC GTA TCT CGC CGA AAG GCT G. The resulting PCR fragment was digested by EcoRI and XhoI and ligated to EcoRI and XhoI digested GFP-CD8a DNA plasmid.

ST6GAL1-GFP (ST-GFP) (*Sun et al., 2020*), CD8a-furin (*Mahajan et al., 2013*), CD8a-CI-M6PR (*Mahajan et al., 2013*) were previously described. RUSH TfR-SBP-GFP was a gift plasmid from J. Bonifacino (*Tie et al., 2017*). GFP-golgin-84 (*Diao et al., 2003*) was a gift plasmid from M. Lowe. Arf1-GFP (*Wessels et al., 2006*) was a gift plasmid from F. van Kuppeveld.

Mouse monoclonal antibody anti-CD8a (OKT8) was from the hybridoma culture supernatant. Mouse monoclonal antibodies against GM130 (#610822) and GS15 (#610960) were purchased from BD Biosciences. Rabbit polyclonal antibody against TGN46 was from Abcam (#ab50595). Alexa Fluor 488, 594, and 647-conjugated goat anti-mouse or anti-rabbit secondary antibodies were purchased from Thermo Fisher Scientific.

## Small molecules

Nocodazole (#487928; working concentration: 33 µM), BFA (working concentration: 5 µM), and cycloheximide (working concentration: 10 µg/ml) were purchased from Merck, Life Technologies Holdings, and Sigma Aldrich, respectively.

## Cell lines

HeLa and 293T cell lines were obtained from the American Type Culture Collection (ATCC). Their identities were authenticated by Short Tandem Repeat analysis (ATCC), and they were routinely screened for mycoplasma contamination using DNA staining.

## Cell culture and transfection

HeLa and 293T cells were maintained in Dulbecco's Modified Eagle's Medium (DMEM) supplemented with 10% fetal bovine serum (FBS) to make it complete DMEM. For cell transfection, Lipofectamine 2000 (Thermo Fisher Scientific) was used per the manufacturer's instructions. For BFA treatment, cells were treated with complete DMEM containing 5 µM BFA for the specified duration. For nocodazole treatment, cells were treated with complete DMEM containing 33 µM nocodazole for 3 hr to induce the formation of Golgi ministacks.

## Immunofluorescence

Cells for immunofluorescence were grown on No. 1.5 Φ12 mm glass coverslips. They were fixed using 4% paraformaldehyde in phosphate-buffered saline (PBS). Following a PBS wash to remove residual paraformaldehyde, any remaining paraformaldehyde within the cells was neutralized with 100 mM NH$_4$Cl. The cells were then processed for immunofluorescence labeling by first incubating with mouse or rabbit primary antibodies, followed by Alexa Fluor 488, 594, and/or 647 conjugated goat anti-mouse or anti-rabbit secondary antibodies. Both primary and secondary antibodies were diluted in PBS containing 5% fetal bovine serum, 2% bovine serum albumin, and 0.1% saponin (Sigma-Aldrich). The labeled cells were mounted in the Mowiol mounting medium, composed of 12% Mowiol 4–88 (EMD Millipore), 30% glycerol, and 100 mM Tris pH 8.5. After the mounting medium had dried, the coverslips were sealed with nail polish and stored at –20 °C.

## Acquiring *LQ*s

To analyze *LQ*s of intra-Golgi transport RUSH reporters, HeLa cells transiently co-expressing individual GFP-tagged RUSH reporter and GalT-mCherry were cultured in complete DMEM supplemented with 16 nM His-tagged streptavidin (in-house purified using Addgene #20860, a gift plasmid from A. Ting) (*Howarth et al., 2006*). Following a 3 hr nocodazole treatment, cells were chased with 50 µM biotin, 10 µg/ml cycloheximide, and 33 µM nocodazole for various durations before fixation.

In another set of experiments to study *LQ*s of Golgi markers under the BFA treatment, nocodazole-treated HeLa cells transiently expressing GalT-mCherry were further incubated with 5 µM BFA and 33 µM nocodazole for various durations before fixation. For these experiments, the Golgi markers were either co-expressed with GalT-mCherry as a GFP-tagged construct or detected as an endogenous protein by immunostaining.

The methodology for acquiring *LQ*s through GLIM has been described in our previous studies (*Tie et al., 2017*; *Tie et al., 2016*). Briefly, cells were further immuno-labeled to visualize endogenous GM130. Ministacks exhibiting fluorescence signals of GM130, transfected GalT-mCherry, and the testing protein were imaged using a wide-field microscope. Ministacks were manually selected for analysis, and fluorescence centers for GM130, GalT-mCherry, and the testing protein were acquired using Fiji (https://imagej.net/software/fiji/). After chromatic aberration correction, coordinates of centers were used to calculate *LQ*s, defined as the ratio of axial distances from GM130 to the testing protein and from GM130 to GalT-mCherry. Intra-Golgi transport kinetic *LQ* data were fitted to the first-order exponential function using OriginPro 2020.

In addition to the manual method mentioned above, we employed a newly developed software tool, Auto-GLIM (https://github.com/Chokyotager/AutoGLIM copy archived at *Lam, 2025*), to automatically analyze our ministack images. To this end, three-color images were acquired as described above. Three z-sections centered around the plane of interest were averaged (average z-projection) for beads and ministacks images. Automated background subtraction was performed using a deep learning segmentation model to first extract the cell contours, followed by a dual annealing optimization algorithm to perform background subtraction to extract the highest number of valid ROIs according to the criteria of GLIM (*Tie et al., 2016*). Final *LQ*s were further subjected to analysis in OriginPro 2020.

## Acquisition of Golgi residence times

The methodology follows protocols previously described (*Sun et al., 2021*; *Sun et al., 2020*). Nocodazole was not used in these experiments. Briefly, HeLa cells on a Φ 35 mm glass-bottom Petri dish were transiently transfected to co-express a GFP-tagged reporter and GalT-mCherry. For RUSH reporters, cells were treated with 50 µM biotin at 20 °C for 2 hr to accumulate the reporter at the

Golgi. Live imaging was performed in a $CO_2$-independent medium (Thermo Fisher Scientific) with 10% FBS, 4 mM glutamine, and 10 µg/ml cycloheximide, using a wide-field microscope until the cellular GFP fluorescence at the Golgi nearly vanished. The resulting time-lapse images were segmented based on GalT-mCherry using Fiji. Total GFP fluorescence within the Golgi was quantified and fitted to the first-order exponential function $y=y0+A1exp(-x/t1)$ in OriginPro 2020. Golgi residence time, $t_{1/2}$, was calculated as $0.693*t1$. We only included time-lapse data with adj. $R^2 \geq 0.80$ and acquisition length $\geq 1.33*t_{1/2}$.

## VHH-anti-GFP-mCherry internalization assay

6x His-tagged VHH-anti-GFP-mCherry was purified as previously described (*Sun et al., 2021*; *Sun et al., 2020*). In the internalization assay, HeLa cells transiently expressing GFP-tagged reporters were continuously incubated with 5 µg/ml VHH-anti-GFP-mCherry at 37 °C for 8 hr. After washing, cells were fixed and imaged.

## Microscopy for GLIM and Golgi residence time

Golgi residence times and most *LQ*s were measured using a wide-field microscope based on Olympus IX83. The microscope featured a ×100 oil objective lens (NA 1.40), a motorized stage for sample positioning, and automated filter cubes to accommodate different fluorescence channels. Dichroic mirrors and filters were optimized for GFP/Alexa Fluor 488, mCherry/Alexa Fluor 594, and Alexa Fluor 647. Imaging was captured with an sCMOS (scientific complementary metal oxide semiconductor) camera (Neo) by Andor. A 200 W metal halide light source (Lumen Pro 200) by Prior Scientific provided illumination. Operational control and data collection were facilitated through Metamorph software by Molecular Devices. The image pixel size is 65 nm. The range of exposure time is 400–5000 ms for each channel.

GS15 *LQ*s during the BFA treatment time course were measured using a spinning disk confocal microscope system comprising Olympus IX81 equipped with a ×100 oil objective lens (NA 1.45), a piezo z stage, Yokogawa CSU-X1 spinning head, 50 mW solid state lasers (488, 561, and 640 nm) (Sapphire; Coherent Inc, Santa Clara, CA, United States), an electron multiplying charge-coupled device (Evolve; Photometrics, Tucson, AZ, United States), and filters optimized for GFP/Alexa fluor 488, mCherry/Alexa Fluor 594, and Alexa Fluor 647. The system was controlled by Metamorph software (Molecular Devices). The image pixel size is 89 nm. The range of exposure time is 200–500 ms for each channel.

## Airyscan microscopy

The Airyscan microscopy was performed using a Zeiss LSM710 confocal microscope, equipped with an Alpha Plan-Apochromat ×100 NA 1.46 objective and the Airyscan module (Carl Zeiss). The system operation was controlled by Zen software (Carl Zeiss). Three lines of laser lights were used: 488, 561, and 640 nm. The emission band was selected to optimize the capture of the emission light while minimizing channel crosstalk. For side averaging, images were acquired under ×63 objective (NA 1.40), zoomed in ×3.5 to achieve 45 nm pixel size using the SR mode. The pixel dwelling time is 1.16 µs. The raw images were processed by Airyscan Zen software.

## Side averaging and en face averaging

En face and side averaging were performed as described previously using Airyscan images (*Tie et al., 2018*; *Tie et al., 2022*). Ministacks with en face views were identified by giantin rings. They were subsequently normalized, expanded, and aligned with the center of the image, followed by averaging using Fiji. The radial mean intensity profile was acquired by Fiji macros (*Tie et al., 2018*). Ministacks with side views were identified by giantin double puncta and subjected to rotation, expansion, and normalization before averaging in Fiji (*Tie et al., 2022*). In a side average image, we define the axial position of a Golgi protein *i* as the *y* component of its center of mass coordinate, $y_i$. In *Figure 3E*, $LQ^{side}$ of SBP-GFP-collagenX is calculated as below.

$$LQ^{side} = \frac{y_{SBP-GFP-collagenX} - y_{GM130}}{y_{GalT-mCherry} - y_{GM130}}$$

$y_{SBP-GFP-collagenX}$, $y_{GM130}$, and $y_{GalT-mCherry}$ are axial positions of SBP-GFP-collagenX, GM130, and GalT-mCherry in their side average images, respectively. $y_{GM130}$ and $y_{GalT-mCherry}$ were measured previously (*Tie et al., 2022*).

In *Figure 6G*, the distance from GM130 to GalT-mCherry, $d_{(GM130-GalT-mCherry)}$, was measured from individual ministacks with side views. The axial line intensity profile of GM130 or GalT-mCherry was subjected to Gaussian fitting in OriginPro2020 (Analysis > Fitting > Non-Linear Curve Fit). The calculated $Xc$ represents the axial position of each protein. $d_{(GM130-GalT-mCherry)}$ of each Golgi ministack is then calculated as

$$d_{GM130-GalT-mCherry} = Xc_{GalT-mCherry} - Xc_{GM130}$$

and subjected to statistical analysis. $d_{(GalT-mCherry-CD8a-furin)}$ in *Figure 6H* was calculated similarly.

## Acknowledgements

We want to thank J Bonifacino (National Institute of Health, USA), F van Kuppeveld (Utrecht University), M Lowe (University of Manchester, UK), F Perez (Institute of Curie, France), and A Ting (Stanford University, USA) for sharing DNA plasmids. This project is supported by the Ministry of Education, Singapore, under its Tier 2 MOE-T2EP30221-0001 and Tier 1 RG 25/22.

---

## Additional information

### Competing interests

Hieng Chiong Tie: is affiliated with Medisix Therapeutics. The author has no other competing interests to declare. Lei Lu: Reviewing editor, eLife. The other authors declare that no competing interests exist.

### Funding

| Funder | Grant reference number | Author |
| --- | --- | --- |
| Ministry of Education - Singapore | Tier 2 MOE-T2EP30221-0001 | Lei Lu |
| Ministry of Education - Singapore | Tier 1 RG 25/22 | Lei Lu |

The funders had no role in study design, data collection and interpretation, or the decision to submit the work for publication.

### Author contributions

Hieng Chiong Tie, Haiyun Wang, Divyanshu Mahajan, Data curation, Formal analysis, Validation, Investigation, Methodology; Hilbert Yuen In Lam, Software, Methodology; Xiuping Sun, Bing Chen, Data curation, Formal analysis; Yuguang Mu, Software, Supervision; Lei Lu, Conceptualization, Resources, Formal analysis, Supervision, Funding acquisition, Investigation, Writing – original draft, Project administration, Writing – review and editing

### Author ORCIDs

Hieng Chiong Tie ⓘ https://orcid.org/0000-0003-2738-8685
Haiyun Wang ⓘ https://orcid.org/0009-0003-5963-2029
Divyanshu Mahajan ⓘ https://orcid.org/0000-0001-5820-6763
Hilbert Yuen In Lam ⓘ https://orcid.org/0000-0001-6129-2703
Lei Lu ⓘ https://orcid.org/0000-0002-8192-1471

Reviewer #1 (Public review): https://doi.org/10.7554/eLife.98582.3.sa1
Reviewer #2 (Public review): https://doi.org/10.7554/eLife.98582.3.sa2
Reviewer #3 (Public review): https://doi.org/10.7554/eLife.98582.3.sa3
Author response https://doi.org/10.7554/eLife.98582.3.sa4

---

## Additional files

**Supplementary files**
MDAR checklist

**Data availability**
The Auto-GLIM software tool has been made available on GitHub (https://github.com/Chokyotager/AutoGLIM copy archived at *Lam, 2025*).

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
