## [Editor Report · eLife Assessment]

This **important** study offers **convincing** evidence that intra-Golgi transport slows from cis to trans and varies between cargos even within the same cisternae, supporting a more stable compartment model. Using nocodazole-induced ministacks, the authors show cargo-specific transport kinetics with distinct velocities and residence times. These findings refine the cisternal progression model and prompt further investigation into alternative mechanisms, such as rapid partitioning or rim progression. This study will be of interest to cell biologists studying membrane trafficking, Golgi organization, and protein secretion, as well as researchers investigating the mechanisms of organelle dynamics and the molecular basis of intracellular transport.

---

## [Referee Report · Reviewer #1 (Public review)]

Summary:

In the manuscript by Tie et.al., the authors couple the methodology which they have developed to measure LQ (localization quotient) of proteins within the Golgi apparatus along with RUSH based cargo release to quantify the speed of different cargos traveling through Golgi stacks in nocodazole induced Golgi ministacks to differentiate between cisternal progression vs stable compartment model of the Golgi apparatus. The debate between cisternal progression model and stable compartment model has been intense and going on for decades and important to understand the basic way of function/organization of the Golgi apparatus. As per the stable compartment model, cisterna are stable structures, and cargo moves along the Golgi apparatus in vesicular carriers. While as per cisternal progression model, Golgi cisterna themselves mature acquiring new identity from the cis face to the trans face and act as transport carriers themselves. In this work, authors provide a missing part regarding intra-Golgi speed for transport of different cargoes as well as the speed of TGN exit and based on the differences in the transport velocities for different cargoes tested favor a stable compartment model. The argument which authors make is that if there is cisternal progression, all the cargoes should have a similar intra-Golgi transport speed which is essentially the rate at which the Golgi cisterna mature. Furthermore, using a combination of BFA and Nocodazole treatments authors show that the compartments remain stable in cells for at least 30-60 minutes after BFA treatment.

Strengths:

The method to accurately measure localization of a protein within the Golgi stack is rigorously tested in the previous publications from the same authors and in combination with pulse chase approaches has been used to quantify transport velocities of cargoes through the Golgi. This is a novel aspect in this paper and differences in intra-Golgi velocities for different cargoes tested makes a case for a stable compartment model.

Weaknesses:

None noted in the revised version of the manuscript.

---

## [Referee Report · Reviewer #2 (Public review)]

Summary:

This manuscript describes the use of quantitative imaging approaches, that have been a key element of the labs work over the past years, to address one of the major unresolved discussions in trafficking: intra-Golgi transport. The approach used has been clearly described in the labs previous papers, and is thus clearly described. The authors clearly address the weaknesses in this manuscript, and do not overstate the conclusions drawn from the data. The only weakness not addressed is the concept of blocking COPI transport with BFA, which is a strong inhibitor and causes general disruption of the system. This is an interesting element of the paper, which I think could be improved upon by using more specific COPI inhibitors instead, although I understand that this is not necessarily straightforward.

I commend the authors on their clear and precise presentation of this body of work, incorporating mathematical modelling with a fundamental question in cell biology. In all, I think that this is a very robust body of work, that provides a sound conclusion in support of the stable compartment model for the Golgi.

General points:

The manuscript contains a lot of background in its results sections, and the authors may wish to consider rebalancing the text: The section beginning at Line 175 is about 90% background and 10% data. Could some data currently in supplementary be included here to redress this balance, or this part combined with another?

Minor points:

Equation 2: A should be in front of the ln2. It's already resolved in equation 3, so likely only needs changing in the text

Line 152: Why is there a lack of experimental data? High ER background and low golgi signal make it difficult to select ministacks: would be good to see examples of these images. Is 0 a relevant timepoint as cargo is still at the ER? Instead would a timepoint <5' be better demonstrate initial arrival in fast cargo, and 0' discarded?

Table 1 Line 474: 1-3 independent replicates: is there a better way of incorporating this into the table to make it more streamlined? It would be useful to see each cargo as a mean with error. Is there a more demonstrative way to present the table, for example (but does not have to be) fastest cargo first (Tintra) as in Table 2?

Line 264 / Fig 3B: It's unclear to me why the VHH-anti-GFP-mCherry internalisation approach was used, when the cells were expressing GFP, that could be used for imaging. Also, this introduces a question over trafficking of the VHH itself, to access the same compartments as the GFP-proteins are localised. It would be useful to describe the choice of this approach briefly in the text.

446 Typo "internalization"

Post-Revision

I thank the authors for their work revising the paper in light of our comments. I am satisfied with their response, and I have no other comments.

---

## [Referee Report · Reviewer #3 (Public review)]

The manuscript by Tie et al. provides a quantitative assessment of intra-Golgi transport of diverse cargos. Quantitative approaches using fluorescence microscopy of RUSH synchronized cargos, namely GLIM and measurement of Golgi residence time, previously developed by the author's team (publications from 20216 to 2022), are being used here.

Most of the results have been already published by the same team in 2016, 2017, 2020 and 2021. In this manuscript, the authors have put together measurement of intra-Golgi transport kinetics and Golgi residence time of many cargos. The quantitative results are supported by a large number of Golgi mini-stacks/cells analyzed. They are discussed with regard to the intra-Golgi transport models being debated in the field, namely the cisternal maturation/progression model and the stable compartments model.

The authors show that different cargos have distinct intra-Golgi transport kinetics and that the Golgi residence time of glycosyltransferases is high. From this and experiment using brefeldinA, the authors suggest that the rim progression model, adapted from the stable compartments model, fits with their experimental data.

Strengths:

The major strength of this manuscript is to put together many quantitative results that the authors previously obtained and to discuss them to advance our understanding of the intra-Golgi transport mechanisms.

The analysis by fluorescence microscopy of intra-Golgi transport is tough and this is a tour de force of the authors even though their approach shows limitations, which are clearly stated. Their work is remarkable in regards of the numbers of Golgi markers and secretory cargos which have been analyzed.

Weaknesses:

Most of the data provided here were already published and thus accessible for the community. The tubular connections between cisternae and the diffusion/biochemical properties of cargos are not taken into account to interpret the results. Indeed, tubular connections and biochemical properties of the cargos may affect their transit through the Golgi and the kinetics with which they reach the TGN for Golgi exit.

The use of nocodazole might affect cellular homeostasis but this is clearly stated by the authors and is acceptable as we need to perturb the system to conduct this analysis.

The manual selection of the Golgi mini-stack being analyzed (where the cargo and the Golgi reference markers are clearly detectable) might introduce a bias in the analysis.

---

## [Author Response]

The following is the authors’ response to the original reviews

**Public Reviews:**

**Reviewer #1 (Public Review):**
Summary:In the manuscript by Tie et.al., the authors couple the methodology which they have developed to measure LQ (localization quotient) of proteins within the Golgi apparatus along with RUSH based cargo release to quantify the speed of different cargos traveling through Golgi stacks in nocodazole induced Golgi ministacks to differentiate between cisternal progression vs stable compartment model of the Golgi apparatus. The debate between cisternal progression model and stable compartment model has been intense and going on for decades and important to understand the basic way of function/organization of the Golgi apparatus. As per the stable compartment model, cisterna are stable structures and cargo moves along the Golgi apparatus in vesicular carriers. While as per cisternal progression model, Golgi cisterna themselves mature acquiring new identity from the cis face to the trans face and act as transport carriers themselves. In this work, authors provide a missing part regarding intra-Golgi speed for transport of different cargoes as well as the speed of TGN exit and based on the differences in the transport velocities for different cargoes tested favor a stable compartment model. The argument which authors make is that if there is cisternal progression, all the cargoes should have a similar intra-Golgi transport speed which is essentially the rate at which the Golgi cisterna mature. Furthermore, using a combination of BFA and Nocodazole treatments authors show that the compartments remain stable in cells for at least 30-60 minutes after BFA treatment.Strengths:The method to accurately measure localization of a protein within the Golgi stack is rigorously tested in the previous publications from the same authors and in combination with pulse chase approaches has been used to quantify transport velocities of cargoes through the Golgi. This is a novel aspect in this paper and differences in intra-Golgi velocities for different cargoes tested makes a case for a stable compartment model.Weaknesses:Experiments are only tested in one cell line (HeLa cells) and predominantly derived from experimental paradigm using RUSH assays where a secretory cargo is released in a wave (not the most physiological condition) and therefore additional approaches would make a more compelling case for the model.

We have added datasets from 293T cells in the revamped manuscript.

**Reviewer #2 (Public Review):**
Summary:This manuscript describes the use of quantitative imaging approaches, which have been a key element of the labs work over the past years, to address one of the major unresolved discussions in trafficking: intra-Golgi transport. The approach used has been clearly described in the labs previous papers, and is thus clearly described. The authors clearly address the weaknesses in this manuscript and do not overstate the conclusions drawn from the data. The only weakness not addressed is the concept of blocking COPI transport with BFA, which is a strong inhibitor and causes general disruption of the system. This is an interesting element of the paper, which I think could be improved upon by using more specific COPI inhibitors instead, although I understand that this is not necessarily straightforward.I commend the authors on their clear and precise presentation of this body of work, incorporating mathematical modelling with a fundamental question in cell biology. In all, I think that this is a very robust body of work, that provides a sound conclusion in support of the stable compartment model for the Golgi.General points:The manuscript contains a lot of background in its results sections, and the authors may wish to consider rebalancing the text: The section beginning at Line 175 is about 90% background and 10% data. Could some data currently in supplementary be included here to redress this balance, or this part combined with another?

In the revamped manuscript, we have moved the background information on rapid partitioning and rim progression models to the Introduction.

**Reviewer #3 (Public Review):**
The manuscript by Tie et al. provides a quantitative assessment of intra-Golgi transport of diverse cargos. Quantitative approaches using fluorescence microscopy of RUSH synchronized cargos, namely GLIM and measurement of Golgi residence time, previously developed by the author's team (publications from 20216 to 2022), are being used here.Most of the results have been already published by the same team in 2016, 2017, 2020 and 2021. In this manuscript, very few new data have been added. The authors have put together measurements of intra-Golgi transport kinetics and Golgi residence time of many cargos. The quantitative results are supported by a large number of Golgi mini-stacks/cells analyzed. They are discussed with regard to the intra-Golgi transport models being debated in the field, namely the cisternal maturation/progression model and the stable compartments model. However, over the past decades, the cisternal progression model has been mostly accepted thanks to many experimental data.The authors show that different cargos have distinct intra-Golgi transport kinetics and that the Golgi residence time of glycosyltransferases is high. From this and the experiment using brefeldinA, the authors suggest that the rim progression model, adapted from the stable compartments model, fits with their experimental data.Strengths:The major strength of this manuscript is to put together many quantitative results that the authors previously obtained and to discuss them to give food for thought about the intraGolgi transport mechanism.The analysis by fluorescence microscopy of intra-Golgi transport is tough and is a tour de force of the authors even if their approach show limitations, which are clearly stated. Their work is remarkable in regards to the numbers of Golgi markers and secretory cargos which have been analyzed.Weaknesses:As previously mentioned, most of the data provided here were already published and thus accessible for the community. Is there is a need to publish them again?The authors' discussion about the intra-Golgi transport model is rather simplistic. In the introduction, there is no mention of the most recent models, namely the rapid partitioning and the rim progression models. To my opinion, the tubular connections between cisternae and the diffusion/biochemical properties of cargos are not enough taken into account to interpret the results. Indeed, tubular connections and biochemical properties of the cargos may affect their transit through the Golgi and the kinetics with which they reach the TGN for Golgi exit.Nocodazole is being used to form Golgi mini-stacks, which are necessary to allow intra-Golgi measurement. The use of nocodazole might affect cellular homeostasis but this is clearly stated by the authors and is acceptable as we need to perturb the system to conduct this analysis. However, the manual selection of the Golgi mini-stack being analyzed raises a major concern. As far as I understood, the authors select the mini-stacks where the cargo and the Golgi reference markers are clearly detectable and separated, which might introduce a bias in the analysis.The terms 'Golgi residence time ' is being used but it corresponds to the residence time in the trans-cisterna only as the cargo has been accumulated in the trans-Golgi thanks to a 20{degree sign}C block. The kinetics of disappearance of the protein of interest is then monitored after 20{degree sign}C to 37{degree sign}C switch.Another concern also lies in the differences that would be introduced by different expression levels of the cargo on the kinetics of their intra-Golgi transport and of their packaging into post-Golgi carriers.

Please see below for our replies to intra-Golgi transport models, the Golgi residence time, and different expression levels of cargos.

**Recommendations for the authors:**

**Reviewer #1 (Recommendations For The Authors):**
The data shown by the authors to measure differential intra Golgi velocities based on previously established methodology make a case for a stable compartment model, however more data is needed to make a complete story and the clarity of presentation can be improved.

We sincerely appreciate the reviewer's insightful, detailed, and constructive feedback. Your thoughtful comments have helped us refine our analyses, clarify key points, and strengthen the overall quality of our manuscript. We are grateful for the time and effort you have dedicated to reviewing our work and providing valuable suggestions. Your input has been instrumental in improving both the scientific rigor and presentation of our findings. Thank you for your thorough and thoughtful review.

Main points:(1) Along with the studies in yeast, which authors describe in this paper, the main evidence for cisternal maturation model in mammalian cells comes from Bonfanti et.al., (https://doi.org/10.1016/S0092-8674(00)81723-7), which used EM to visualize a wave of Collagen through Golgi stacks. It is therefore important this work needs to include collagen as one of the cargos tested. Can the authors use the RUSH-Col1AGFP (see: https://doi.org/10.1083/jcb.202005166) as a cargo to monitor intra-Golgi velocities?I understand that Hela cells are not professional collagen-secreting, but the authors can use U2OS cells to measure collagen export and two other extreme (slow and fast) cargos to validate the same trend in intra-Golgi transport velocities is seen in other cell lines. This will address three concerns: a. This is not a Hela-specific phenomenon; b. Transport of large cargoes like collagen agree with their proposal; c. To see if the same cargo has the same (similar) intra-Golgi speed and the trend between different cargoes is conserved across cell lines.

Due to the difficulty of manipulating and imaging the procollagen-I RUSH reporter, we selected the collagenX-RUSH reporter (SBP-GFP-collagenX) instead. Our previous study (Tie et al., eLife, 2028) demonstrated that SBP-GFP-collagenX assembles as a large molecular weight particle, each having ~ 190 copies of SBP-GFP-collagenX. With an estimated mean size of ~ 40 nm, these aggregates are not as large as FM4 aggregates and procollagen-I (> 300 nm) and, therefore, are not excluded from conventional transport vesicles, which typically have a size of 50 – 100 nm. However, collagenX has distinct intra-Golgi transport behaviour from conventional secretory cargos -- while conventional secretory cargos localize to the cisternal interior, collagenX partitions to the cisternal rim (Tie et al., eLife, 2028).

We studied the intra-Golgi transport of SBP-GFP-collagenX in HeLa cells via GLIM and side averaging. The new results are included in Figure 3 of the revamped manuscript. CollagenX has similar intra-Golgi transport kinetics as conventional secretory cargos, displaying the first-order exponential function in *LQ* vs. time and velocity vs. time plots.

The side-averaging images are consistent with previous and current results. collagenX displays a double-punctum during the intra-Golgi transport, indicating a cisternal rim localization, as expected for large secretory cargos. Therefore, our new data demonstrated that cisternal rim partitioned large-size secretory cargos might follow intra-Golgi transport kinetics similar to those of cisternal interior partitioned conventional secretory cargos.

We tried SBP-GFP-CD59 and SBP-GFP-Tac-TC, cargos with fast and slow intra-Golgi transport velocities, respectively, in 293T cells. Results are included in Figure 2, Supplementary Figure 2, and Table 1 of the revamped manuscript. We found that SBP-GFPTac-TC showed similar *t*_intra_s, 17 and 14 min, respectively, in HeLa and 293T cells. Considering our previous finding that glycosylation has an essential role in the Golgi exit (Sun et al., JBC, 2020), the distinct intra-Golgi transport kinetics of SBP-GFP-CD59 (*t*_intra_s, 13 and 5 min, respectively, in HeLa and 293T cells) might be due to its distinct luminal glycosylation between HeLa and 293T cells. Supporting this hypothesis, SBP-GFP-Tac-TC does not have any glycosylation sites due to the truncation of the Tac luminal domain.

(2) RUSH assay has its own caveats which authors also refer to in the manuscript. Authors should test their model by using pulse chase approaches by SNAP tagged constructs which will allow them to do pulse chase assays without the requirement to release cargo as a wave (see: doi: 10.1242/jcs.231373). It is not necessary to test all the cargoes but the two on the ends of the spectrum (slow and fast). To avoid massive overexpression, authors could express the proteins using weaker promoters. Authors could also use this approach to simultaneously measure the two cargoes by tagging them with CLIP and SNAP tags and doing the pulse chase simultaneously (see: DOI: 10.1083/jcb.202206132). In this case it may be difficult to stain both GM130 and TGN, but authors could monitor the rate of segregation from the GM130 signal.

During the RUSH assay, the sudden release of a large amount of secretory reporters does not occur under native secretory conditions and, consequently, might introduce artifacts. The reviewer suggests using pulse-chase labeling of SNAP (or CLIP)-tagged secretory cargos, which occurs in a steady state and hence more closely resembles native secretory transport. This is an excellent suggestion. However, we have not yet tested this method due to the following concerns.

The standard protocol involves blocking existing reporters, pulse-labeling newly synthesized reporters, and chasing their movement along the secretory pathway. However, the typical 20minute pulse labeling period used in the two references would be too long, as a substantial portion of the reporters would already reach the *trans*-Golgi or exit the Golgi before the chase begins. Conversely, reducing the pulse labeling time would significantly weaken the GLIM signal.

(3) While the intra-Golgi velocities are different for different cargoes tested, authors should show a control that the arrival of the cargoes from ER to the cis-Golgi follows similar kinetics or if there are differences there is no correlation with the intra-Golgi velocities. In other words, do cargoes which show slow intra-Golgi velocities also take more time to reach the cis-Golgi and vice versa.

In nocodazole-induced Golgi ministacks, the ER exit site, ERGIC, and *cis*-Golgi are spatially closely associated. At the earliest measurable time point—5 minutes after biotin treatment— we observed that the secretory cargo had already reached the *cis*-Golgi (Figure 2 and Supplementary Figure 2). The rapid ER-to-*cis*-Golgi transport exceeds the temporal resolution of our current protocol, making it difficult to address the reviewer’s question (see our reply to Minor Points (2) of Reviewer #2 for more detailed discussion on this).

(4) Were the different cargos traveling (at different speeds) through Golgi at the rims, or in the middle of ministack, or by vesicles?

Please also refer to our reply to Question 1 of Reviewer #1. For the nocodazole-induced Golgi ministack, we previously investigated the lateral cisternal localization of RUSH secretory reporters using our en face average imaging (Tie et al., eLife, 2018). We found that small or conventional cargos (such as CD59 and E-cadherin) partition to the cisternal interior while large cargos (collagenX and FM4-CD8a) partition to the cisternal rim during their intra-Golgi transport. Using GLIM, we showed that the intra-Golgi transport kinetics of collagenX is similar to that of small cargos as both follow the first-order exponential function (Figure 3A-C). Therefore, cisternal rim partitioned large size secretory cargos might have intra-Golgi transport kinetics similar to those of cisternal interior partitioned conventional secretory cargos.

(5) Figure 4, under both nocodazole and BFA treatment for 30mins, would the stacks have the same number (274 nm per LQ) as thickness? Or does it shrink a little? Considering extended BFA treatment reduced intact Golgi ministacks. This is important to understand the LQ numbers of those Golgi proteins. Besides, can they include one ERGIC marker in this assay, would it be approaching cis-Golgi? Images used for quantification in Figure 4 should be shown in the main figure.

We define the axial size of the Golgi ministack as the axial distance from the GM130 to the GalT-mCherry, *d*_(GM130-GalT-mCherry)_, measured using the Gaussian centers of their line intensity profiles. As the reviewer suggested, we measured the axial size of the ministack during the nocodazole and BFA treatment. Indeed, we found a decrease in the ministack axial size from 300 ± 10 nm at 0 min to 190 ± 30 nm at 30 min of BFA treatment. This observation is further confirmed by our side average imaging. The new data is presented in Fig. 6G.

Our study focuses on changes in the organization of the Golgi ministack. So, we didn’t include ERGIC53 in the current analysis. Instead, we quantified the axial distance between GalTmCherry and CD8a-furin, *d*_(GalT-mCherry-CD8a-furin)_, and found that it decreased from 200 ± 20 nm at 0 min to 100 ± 30 nm at 30 min of BFA treatment, suggesting the collapse of the TGN. The collapse of the TGN is further visualized by our side average imaging. The new data is presented in Fig. 6H.

Therefore, our new data demonstrates that the Golgi ministack shrinks, and the TGN collapses under BFA treatment.

Minor points:(1) The LQ data come from confocal/airy scan images, but no such images were shown in this paper. The authors can't assume every reader to have prior knowledge of their previous work. It will be beneficial to have one example image and how the LQ was measured.

As advised by the reviewer, we have prepared Supplementary Figure 1 to provide a brief illustration of the principle behind GLIM and image processing steps involved.

(2) The cargos used in this paper need to be introduced: what are they, how were they used in previous literature. Especially the furin constructs come out of the blue (also see point 7).

As suggested by the reviewer, we have included a schematic diagram in Fig. 1 of the revised manuscript to illustrate all RUSH reporters and their corresponding ER hooks. In this diagram, we also highlight the key sequence differences in the cytosolic tails of different furin mutants.

Additionally, we have added references for each RUSH reporter at the beginning of the Results and Discussion section.

(3) There are two categories of exocytosis, constitutive and regulated. It important to state that the phenomenon observed is in cells predominantly showing only constitutive secretion.

As the reviewer advised, we have added the following sentences in the section titled “Limitations of the study”.

“Third, all RUSH reporters used in this study are constitutive secretory cargos. As a result, the intra-Golgi transport dynamics observed here might not reflect those of regulated secretion, which involves the synchronized release of a large quantity of cargo in response to a specific signal.”

(4) All the cargoes show a progressive reduction in instantaneous velocities from cis to medial to trans. Authors should discuss how do they mechanistically explain this. Is the rate of vesicle production progressively decreasing from cis to trans and if so, why?

As our imaging methods cannot differentiate vesicles from the cisternal rim, we could not tell if the vesicle production rate had changed during the intra-Golgi transport. We have provided an explanation of the progressive reduction of the intra-Golgi transport velocity in the Results and Discussion section. Please see the text below.

“The progressive reduction in intra-Golgi transport of secretory cargo might result from the enzyme matrix's retention at the trans-Golgi. As the secretory cargos progress along the Golgi stack from the cis to the trans-side, more and more cargos become temporarily retained in the trans-Golgi region, gradually reducing their overall intra-Golgi transport velocity. If the release or Golgi exit of these cargos from the enzyme matrix follows a constant probability per unit time, i.e., a first-order kinetics process, the rate of cargo exiting from the Golgi should follow the first-order exponential function. Since the mechanism underlying intra-Golgi transport kinetics reflects fundamental molecular and cellular processes of the Golgi, further experimental data are essential to rigorously test this hypothesis.”

(5) The supp file 1 nicely listed the raw data for plotting, and n for numbers of ministacks. Could the authors also show number of cells or experiment repeats?

In the revamped version of the Supplementary File 1, we have added the cell number for each *LQ* measurement.

(6) This recent work used novel multiplexing methods to show that nocodazole-treated cells had similar protein organization as in control may be cited. It also showed the effect of BFA. https://www.cell.com/cell/abstract/S0092-8674(24)00236-8.

We have added this reference to the Introduction section to support that nocodazole-induced Golgi ministacks have a similar organization as the native Golgi. However, our BFA treatment was combined with the nocodazole treatment, while this paper’s BFA treatment does not contain nocodazole.

(7) Figure 1G-J, authors should show a schematic to show the difference between different furin constructs. Also, LQ values in Fig 1I start from 1. Authors may need to include even earlier timepoints.

As suggested by the reviewer, we have shown the domain organization of wild type and mutant furin RUSH reporters in Figure 1, highlighting key amino acids in the cytosolic tail. Please also see our reply to Minor Points (2) of Reviewer #1.

In the revised manuscript, Fig. 1l (SBP-GFP-CD8a-furin-AC #1) has been updated to become Fig. 2J. In this dataset, the first time point was selected at a relatively late stage (20 min), resulting in an initial *LQ* value of 0.92. However, this should not pose an issue, as SBP-GFPCD8a-furin-AC reaches a plateau of ~ 1.6. The number of data points is sufficient to capture the rising phase and fit the first-order exponential function curve with an adjusted *R*^2^ = 0.99. Furthermore, we have four independent datasets in total on the intra-Golgi transport of SBPGFP-CD8a-furin-AC (#1-4), demonstrating the consistency of our measurements.

(8) Figure 2A need to show the data points, not just the lines.

In the revamped manuscript, Fig. 2A has been updated to become Fig. 4A. The plot of Fig. 4A is calculated based on Equation 3.

\begin{document}$\frac{d L Q}{d t}=\frac{\ln 2}{t_{\text {intra }}}\left(y_{0}-L Q\right)$\end{document}

So, it does not have data points. However, *t*_intra_ is calculated based on the experimental *LQ* vs. *t* kinetic data.

(9) Imaging and camera settings like exposure time, pixel size, etc should be reported in Methods.

As suggested by the reviewer, we have supplied this information in the Materials and Methods section of the revised manuscript.

(1) The exposure time and pixel size for the wide-field microscopy:

“The image pixel size is 65 nm. The range of exposure time is 400 – 5000 ms for each channel.”

(2) The exposure time and pixel size for the spinning disk confocal microscopy: “The image pixel size is 89 nm. The range of exposure time is 200 – 500 ms for each channel.”

(3) The pixel dwelling time and pixel size for the Airyscan microscopy:

“For side averaging, images were acquired under 63× objective (NA 1.40), zoomed in 3.5× to achieve 45 nm pixel size using the SR mode. The pixel dwelling time is 1.16 µs.”

**Reviewer #2 (Recommendations For The Authors):**

We sincerely appreciate the reviewer's insightful, detailed, and constructive feedback. Your thoughtful comments have helped us refine our analyses, clarify key points, and strengthen the overall quality of our manuscript. We are grateful for the time and effort you have dedicated to reviewing our work and providing valuable suggestions. Your input has been instrumental in improving both the scientific rigor and presentation of our findings. Thank you for your thorough and thoughtful review.

Minor points:(1) Equation 2: A should be in front of the ln2. It's already resolved in equation 3, so likely only needs changing in the text

As suggested by the reviewer, we have changed it accordingly.

\begin{document}$\frac{d L Q}{d t}=\frac{A \ln 2}{t_{\text {intra }}} e^{\left(-\frac{l n 2}{t_{\text {intra }}} t\right)}$\end{document}

(2) Line 152: Why is there a lack of experimental data? High ER background and low golgi signal make it difficult to select ministacks: would be good to see examples of these images. Is 0 a relevant timepoint as cargo is still at the ER? Instead would a timepoint <5' be better demonstrate initial arrival in fast cargo, and 0' discarded?

We observed that RUSH reporters typically do not exit the ER in < 5 min of biotin treatment, resulting in a high ER background and low Golgi signal. Example images of SBP-GFP-CD59 are shown below (scale bar: 10 µm). Possible reasons include: (1) the time required for biotin diffusion into the ER, (2) the time needed to displace the RUSH hook from the RUSH reporter, and (3) the time for recruitment of RUSH reporters to ER exit sites. As a result, we could not obtain *LQ*s for time points earlier than 5 min during the biotin chase.

**Author response image 1. sa4fig1:** 

Despite the challenge in measuring *LQ*s at early time points, 0 is still a relevant time point. At *t* = 0 min, RUSH reporters should be at the ER membrane near the ER exit site, a definitive pre-Golgi location along the Golgi axis, although we still don’t have a good method to determine its *LQ*.

(3) Table 1 Line 474: 1-3 independent replicates: is there a better way of incorporating this into the table to make it more streamlined? It would be useful to see each cargo as a mean with error. Is there a more demonstrative way to present the table, for example (but does not have to be) fastest cargo first (Tintra) as in Table 2?

As suggested by the reviewer, we revised Table 1. We calculated the mean and SD of *t*_intra_ and arranged our RUSH reporters in ascending order based on their *t*_intra_ values.

(4) Line 264 / Fig 3B: It's unclear to me why the VHH-anti-GFP-mCherry internalisation approach was used, when the cells were expressing GFP, that could be used for imaging. Also, this introduces a question over trafficking of the VHH itself, to access the same compartments as the GFP-proteins are localised. It would be useful to describe the choice of this approach briefly in the text.

Here, the surface-labeling approach is used to investigate if GFP-Tac-TC possesses a Golgi retrieval pathway after its exocytosis to the plasma membrane. When VHH-anti-GFP-mCherry is added to the tissue culture medium, it binds to the cell surface-exposed GFP-fused MGAT1, MGAT2, Tac, Tac-TC, CD8a, and CD8a-TC. Next, VHH-anti-GFP-mCherry traces the internalized GFP-fused transmembrane proteins. The surface-labeling approach has two advantages in this case. (1) It is much more sensitive in revealing the minor number of GFPtransmembrane proteins at the plasma membrane and endosomes, which are usually drowned in the strong Golgi and ER background fluorescence in the GFP channel. (2) While the GFP fluorescence distribution has reached a dynamic equilibrium, the surface labeling approach can reveal the endocytic trafficking route and dynamics.

As the reviewer suggested, we added the following sentence to describe the choice of the cellsurface labeling – “By binding to the cell surface-exposed GFP, VHH-anti-GFP-mCherry serves as a sensitive probe to track the endocytic trafficking itinerary of the above GFP-fused transmembrane proteins”.

Regarding the trafficking of VHH-anti-GFP-mCherry itself, in HeLa cells that do not express GFP-fused transmembrane proteins, VHH-anti-GFP-mCherry can be internalized by fluidphase endocytosis. However, the fluid-phase endocytosis is negligible under our experimental condition, as we previously demonstrated (Sun et al., JCS, 2021; PMID: 34533190).

(5) 446 Typo "internalization"

It has been corrected.

**Reviewer #3 (Recommendations For The Authors):**
Below are my recommendations for the authors to improve their manuscript:

We sincerely appreciate the reviewer's insightful, detailed, and constructive feedback. Your thoughtful comments have helped us refine our analyses, clarify key points, and strengthen the overall quality of our manuscript. We are grateful for the time and effort you have dedicated to reviewing our work and providing valuable suggestions. Your input has been instrumental in improving both the scientific rigor and presentation of our findings. Thank you for your thorough and thoughtful review.

(1) Line 48: Tie at al. 2016 is cited. Please add references to original work showing that cargos transit from cis to trans Golgi cisternae.

After reviewing the literature, we identified two references that provide some of the earliest morphological evidence of secretory cargo transit from the *cis*- to the *trans*-Golgi:

(1) Castle et al, JCB, 1972; PMID: 5025103

(2) Bergmann and Singer, JCB, 1983; PMID: 6315743

The first study utilized pulse-chase autoradiographic EM imaging to track secretory protein movement, while the second employed immuno-EM imaging to observe the synchronized release of VSVGtsO45. Accordingly, we have removed Tie et al., 2016 and replaced it with these newly identified references.

(2) I would suggest to cite earlier (in the Introduction) the rapid partitioning and rim progression models.

As suggested, we have moved the rapid partitioning and rim progression models to the Introduction section.

(3) Figure 1: LQ vs. time plot for SBP-GFP-CD8a-furinAC (panel I, 0.9 to 1.75 in 150 min) is different from Fig 7G of Tie et al. 2016 (LQ O-1.5 in 100 min). Please comment on why those 2 sets of data are different.

We appreciate the reviewer for pointing out this error. In our previous publication (Tie et al., MBoC, 2016), we presented a total of four datasets on SBP-GFP-CD8a-furin-AC. However, in the earlier version of our manuscript, we mistakenly listed only three datasets, inadvertently omitting Fig. 7G from Tie et al., MBoC, 2016.

In the revised version, we have now included Fig. S2T (SBP-GFP-CD8a-furin-AC #4), which corresponds to Fig. 7G from Tie et al., MBoC, 2016.

(4) As mentioned in the public review, I think measurement of the expression level of the cargos is necessary to compare their transport kinetics.

The reviewer raises a valid concern that is challenging to address. All our data were obtained by imaging overexpressed reporters, and we assume that their overexpression does not significantly impact the Golgi or the secretory pathway. Our previous studies have demonstrated that overexpression does not substantially affect *LQ*s (Figure S2 of Tie et al., MBoC, 2016, and Figure S1 of Tie et al., JCB, 2022).

We acknowledge this concern as one of the limitations in our study at the end of our manuscript:

“First, our approach relied on the overexpression of fluorescence protein-tagged cargos. The synchronized release of a large amount of cargo could significantly saturate and skew the intra-Golgi transport.”

(5) To my opinion, cisternal continuities would also affect retrograde transport (accelerate) (by diffusion for instance) and not only retrograde transport. Please comment on how this would affect intra-Golgi transport kinetics.

We believe the reviewer is suggesting “cisternal continuities would also affect retrograde transport (accelerate) (by diffusion for instance) and not only anterograde transport.”

Transient cisternal continuities have been reported to facilitate the anterograde transport of large quantities of secretory cargos (Beznoussenko et al., 2014; PMID: 24867214) (Marsh et al., 2004; PMID: 15064406) (Trucco et al., 2004; PMID: 15502824). However, we are not aware of any reports demonstrating that such continuities facilitate the retrograde transport of secretory cargo, although Trucco et al. (2004) speculated that Golgi enzymes might use these connections to diffuse bidirectionally (anterograde and retrograde direction). For this reason, we did not discuss this scenario in our manuscript.

(6) Lines 188-190: I don't understand why the rapid partitioning model is excluded. Please detail more the arguments used for this statement.

Below is the section from the Introduction that addresses the reviewer's question.

“This model (rapid partitioning model) suggests that cargos rapidly diffuse throughout the Golgi stack, segregating into multiple post-translational processing and export domains, where cargos are packed into carriers bound for the plasma membrane. Nonetheless, synchronized traffic waves have been observed through various techniques, including EM (Trucco et al., 2004) and advanced light microscopy methods we developed, such as GLIM and side-averaging(Tie et al., 2016; Tie et al., 2022). These findings suggest that the rapid partitioning model might not accurately represent the true nature of the intra-Golgi transport.”

(7) I would suggest replacing the 'Golgi residence time' by another name as it reflects mainly the time of Golgi exit if I am not mistaken.

We believe the term “Golgi residence time” more accurately reflects the underlying mechanism – retention. The same approach to measure the Golgi residence time can also be applied to Golgi enzymes such as ST6GAL1. Its slow Golgi exit kinetics (*t*_1/2_ = 5.3 hours) (Sun et al., JCS, 2021) should be primarily due to a strong Golgi retention at its steady state Golgi localization.

In contrast, the conventional secretory cargos’ Golgi exit times are usually much shorter (*t*_1/2_ < 20 min) (Table 2) due to weaker Golgi retention. In a broader sense, the Golgi exit kinetics of a secretory cargo should be influenced by its Golgi retention. Furthermore, we have consistently used the term “Golgi residence time” in our previous publications. So, we propose maintaining this terminology in the current manuscript.

(8) Lines 300-306: I would suggest that the authors remove this part as it is highly speculative and not supported by data.

We have relocated this discussion to the section titled "Our data supports the rim progression model, a modified version of the stable compartment model."

Our enzyme matrix hypothesis offers a potential explanation for key observations, including the differential cisternal localization of small and large cargos and the interior localization of Golgi enzymes. Cryo-FIB-ET has shown that the interior of Golgi cisternae is enriched with densely packed Golgi enzymes (Engel et al., PNAS, 2015; PMID: 26311849), supporting this hypothesis.

Additionally, this hypothesis helps explain the gradual reduction in intra-Golgi transport velocities of secretory cargos, as requested by Reviewer #1 (Minor Points 4). For these reasons, we propose retaining this discussion in the manuscript.

(9) In Figure 3B, percentage of MGAT2-GFP cells with anti-GFP signal at the Golgi is of 41% while Sun et al. 2021 reported 25%, please comment this difference. Reply:

We included more cells for the quantification. The percentage of cells showing Golgi localization of VHH-anti-GFP-mCherry is now 32% (*n* = 266 cells). The observed difference, 32% vs. 25% (Sun et al., JCS, 2021), is likely due to uncontrollable variations in experimental conditions, which might have influenced the endocytic Golgi targeting efficiency.

(10) The effects of brefeldinA are pleiotropic as it disassembles COPI and clathrin coats but also induces tubulation of endosomes. I would recommend using Golgicide A, which is more specific.

We agree with the reviewer that Golgicide A might be more specific as an inhibitor of Arf1. We will certainly consider using this inhibitor next time.